# Dual-specific autophosphorylation of kinase IKK2 enables phosphorylation of substrate IκBα through a phosphoenzyme intermediate

Prateeka Borar[1], Tapan Biswas[2†], Ankur Chaudhuri[3,4†], Pallavi T Rao[4,5], Swasti Raychaudhuri[4,5], Tom Huxford[6], Saikat Chakrabarti[3,4], Gourisankar Ghosh[2], Smarajit Polley[1]*

[1]Department of Biological Sciences, Bose Institute, Kolkata, India; [2]Department of Chemistry and Biochemistry, University of California San Diego, La Jolla, United States; [3]Structural Biology and Bioinformatics Division, CSIR-Indian Institute of Chemical Biology, Kolkata, India; [4]Academy of Scientific and Innovative Research (AcSIR), Ghaziabad, India; [5]CSIR-Centre for Cellular and Molecular Biology, Uppal Road, Hyderabad, India; [6]Department of Chemistry and Biochemistry, San Diego State University, San Diego, United States

*For correspondence: smarajit.polley@gmail.com; spolley@jcbose.ac.in

†These authors contributed equally to this work

Competing interest: The authors declare that no competing interests exist.

## eLife Assessment

This study presents **fundamental** findings that could redefine the specificity and mechanism of action of the well-studied Ser/Thr kinase IKK2 (a subunit of inhibitor of nuclear factor kappa-B kinase (IKK) that propagates cellular response to inflammation). **Solid** evidence supports the claim that IKK2 exhibits dual specificity that allows tyrosine autophosphorylation and the authors further show that auto-phosphorylated IKK2 is involved in an unanticipated relay mechanism that transfers phosphate from an IKK2 tyrosine onto the IκBα substrate. The findings are a starting point for follow-up studies to confirm the unexpected mechanism and further pursue functional significance.

**Abstract** Rapid and high-fidelity phosphorylation of serine residues at positions 32 and 36 of IκBα by IKK2/β, a highly conserved prototypical Ser/Thr kinase in vertebrates, is critical for canonical NF-κB activation. Here, we report that human IKK2 not only phosphorylates substrate serine residues and autophosphorylates its own activation loop, but also autophosphorylates at a tyrosine residue proximal to the active site and is, therefore, a dual-specificity kinase. We observed that mutation of Y169, an autophosphorylatable tyrosine located at the DFG +1 (DLG in IKK1/α and 2) position, to phenylalanine renders IKK2 incapable of catalyzing phosphorylation at S32 within its IκBα substrate. We also observed that mutation of the phylogenetically conserved ATP-contacting residue K44 in IKK2 to methionine converts IKK2 to an enzyme that no longer catalyzes specific phosphorylation of IκBα at S32 or S36, but rather directs phosphorylation of IκBα at other residues. Lastly, we report evidence of a phospho-relay from autophosphorylated IKK2 to IκBα in the presence of ADP. These observations suggest an unusual evolution of IKK2, in which autophosphorylation of tyrosine(s) in the activation loop and the conserved ATP-contacting K44 residue provide its signal-responsive substrate specificity and ensure fidelity during NF-κB activation.

## Introduction

Protein kinases confer novel identity and, often, novel functionality to their substrates by adding phosphate moieties to specific sites, thus playing key regulatory roles in a diverse array of signaling systems. One such example is the action of Inhibitor of κB Kinase-complex (IKK-complex), which is composed of two catalytic subunits, IKK1 (also known as IKKα) and IKK2 (or IKKβ), and an essential regulatory scaffolding protein, NEMO (IKKγ). IKK-complex precisely triggers induction of NF-κB family transcription factors in metazoans in response to inflammatory or pathogenic signals (*Hoffmann and Baltimore, 2006*; *DiDonato et al., 1997*; *Rothwarf et al., 1998*; *Zandi et al., 1997*). In resting cells, the activity of IKK2 is maintained at a low basal level (*Ghosh and Karin, 2002*; *Häcker and Karin, 2006*; *Hinz and Scheidereit, 2014*; *Liu et al., 2012*). In response to inflammatory or pathogenic signaling cues, the IKK2 subunit becomes activated via phosphorylation at two serine residues within its activation loop and subsequently phosphorylates two critical serine residues (S32 and S36) near the N-terminus of IκBα (Inhibitor of NF-κB α), marking the inhibitor protein for ubiquitin-dependent 26 *S* proteasome-mediated degradation (*Figure 1A*). Signal-responsive phosphorylation of IκBα refers to phosphorylation of its S32 and S36 residues by the IKK activity. Degradation of IκBα liberates NF-κB, which then translocates to the nucleus to execute its gene expression program (*Hayden and Ghosh, 2008*; *Hinz and Scheidereit, 2014*; *Karin and Ben-Neriah, 2000*; *Scheidereit, 2006*). Mutation of IκBα residues S32 and S36 to phospho-ablative alanine converts it into a super-repressor of NF-κB that is incapable of being degraded in response to NF-κB signaling (*Brown et al., 1995*; *Lin et al., 1995*), highlighting the need for exquisite specificity of IKK2 in phosphorylating these two serines (*DiDonato et al., 1997*). The IKK2 subunit fails to phosphorylate IκBα when S32 and S36 are mutated even to threonine, another phosphorylatable residue. Despite its critical role in the regulation of NF-κB via high-fidelity phosphorylation specificity toward IκBα, IKK2 behaves with functional pleiotropy toward other substrates and in other contexts (*Antonia et al., 2021*; *Schröfelbauer et al., 2012*; *Schröfelbauer and Hoffmann, 2011*).

IKK2 is a multidomain protein consisting of a kinase domain (KD) in association with a ubiquitin-like domain (ULD) followed in sequence by a scaffold dimerization domain (SDD) and a C-terminal region that contains the NEMO-binding domain (NBD; *Figure 1B*). As mentioned previously, the hitherto enigmatic activation of IKK2 from its inactive state is manifested in the phosphorylation status of two serine residues, S177 and S181 (S176 and S180 for IKK1), located within the activation loop (AL) of the KD (*Huse and Kuriyan, 2002*; *Liu et al., 2013*; *Polley et al., 2013*; *Xu et al., 2011*). Phospho-mimetic substitution of these two serines to glutamates (S177E, S181E; henceforth EE) renders IKK2 constitutively active. It has been demonstrated that the upstream kinase TAK1 primes IKK2 through phosphorylation at S177, leading to its autophosphorylation at S181 and full activation in cells (*Zhang et al., 2014*). Alternatively, it has also been shown that increased oligomerization upon association with NEMO and linear or Lys63-linked poly-ubiquitin chains, or due to a high concentration of IKK2, enables *trans* autophosphorylation of IKK2 at S177 and S181 (*Chen, 2012*; *Du et al., 2022*; *Ea et al., 2006*; *Polley et al., 2013*). NEMO appears not only to aid in the activation of the IKK complex, but also to direct IKK activity toward IκBα bound to NF-κB (NF-κB:IκBα complex; *Schröfelbauer et al., 2012*). The SDD of IKK2 and the C-terminal segment (residues 410–756, collectively) are also required for specific phosphorylation of IκBα at S32 and S36. A shortened version of IKK2 with deletion of these regions failed to retain the above-mentioned exquisite specificity of IKK2 but phosphorylated IκBα at serine and threonine residues within its C-terminal PEST (residues 281–302) region (*Shaul et al., 2008*). Interestingly, this C-terminal PEST region of IκBα, which is rich in proline, glutamic acid, serine, and threonine residues, is phosphorylated in vivo by other kinases that are not capable of phosphorylating IκBα at S32 and S36 (*Barroga et al., 1995*; *Tergaonkar et al., 2003*). Despite its central role in the regulation of transcription factor NF-κB, the connection or interdependence of IκBα phosphorylation at the PEST region and the signal-responsive S32/S36 sites remains elusive.

In this study, we employed structure-based biochemical and cell-based experimental approaches to explore the basis of IKK2 specificity toward S32/S36 of IκBα that is required for rapid induction of NF-κB transcriptional activity in response to canonical signaling. Biochemical analysis of catalytically competent IKK2 revealed that it is capable of autophosphorylation at tyrosine residues in addition to autophosphorylation of its own serines or phosphorylation of serines in its substrate IκBα. Therefore, IKK2 is a dual-specificity protein kinase. IKK2 tyrosine autophosphorylation depends upon prior phosphorylation of activation loop S177/S181 and plays a significant role in phosphorylation

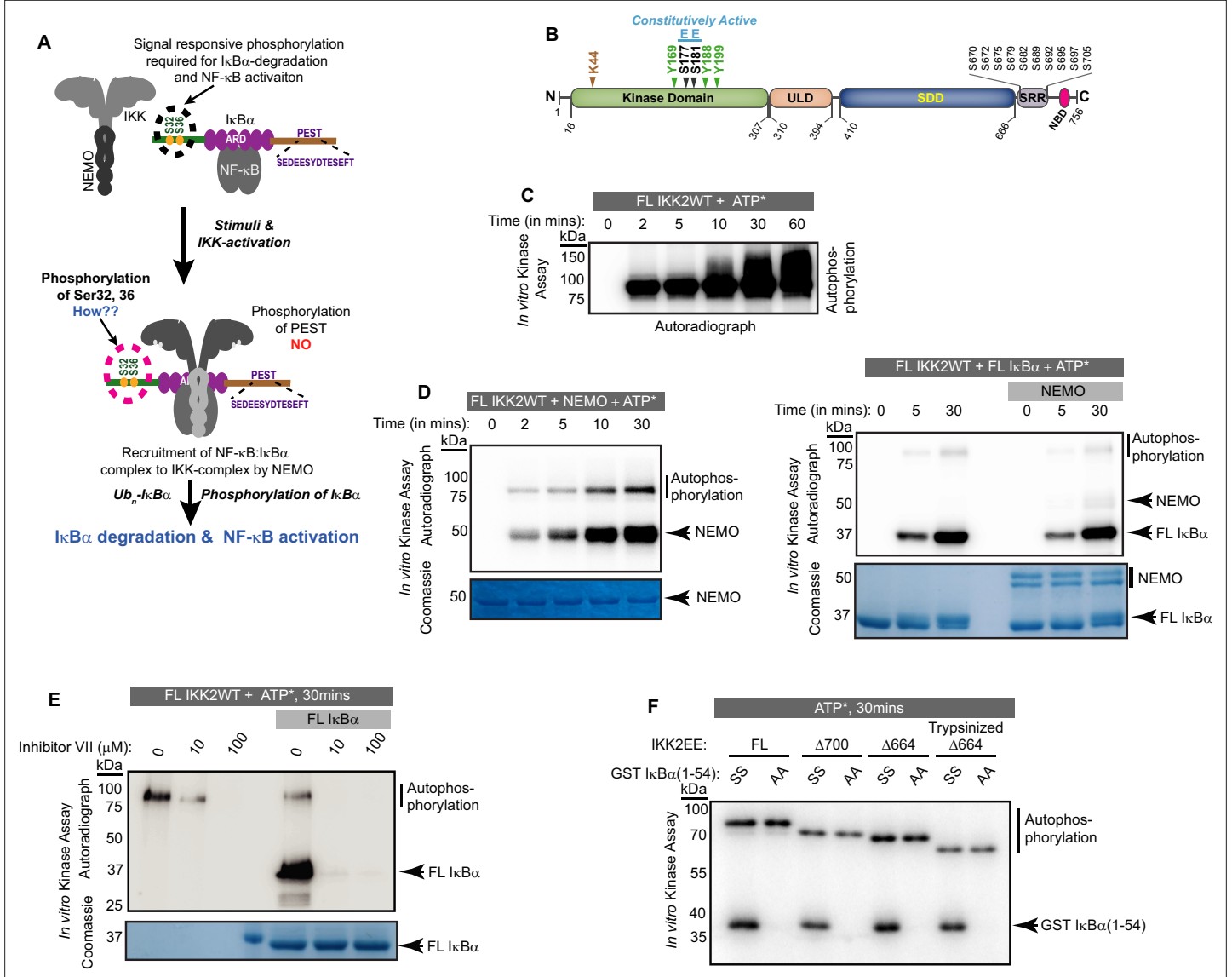

**Figure 1.** Autophosphorylation of IKK2 at hitherto uncharacterized sites. (**A**) A schematic of the IKK-complex (pre- and post-stimulation) showing binding of NEMO helping activation of IKK as well as channelizing its recognition of substrate IκBα. The mechanism of specific phosphorylation at Ser32 and Ser36 of IκBα is unclear (**B**) Domain organization of IKK2 based on the X-ray structures highlighting its functional kinase domain (KD), ubiquitin-like domain (ULD), scaffold dimerization domain (SDD), and NEMO-binding domain. Serine residues in the activation loop - substitution of which to glutamate renders IKK2 constitutively active, and those in the SRR region known to be phosphorylated are marked. Tyrosine residues in the activation loop and the conserved ATP-interacting Lys44 are also marked. (**C**) In vitro kinase assay showing autophosphorylation of wild-type full-length IKK2 (FL IKK2WT) upon incubation with γ³²P radiolabeled ATP for different time periods. (**D**) Similar in vitro kinase assays performed to assess the effect of NEMO on autophosphorylation of FL IKK2WT (left panel), and the effect of NEMO and IκBα on autophosphorylation and substrate phosphorylation (right panel) activities of FL IKK2WT. (**E**) In vitro kinase assay (schematic depicted in *Figure 1—figure supplement 1B*) showing the effect of different concentrations of the Inhibitor VII on FL IKK2WT autophosphorylation and IκBα substrate phosphorylation (this assay was performed twice). (**F**) Kinase assay with radiolabeled ATP displaying auto- and substrate-phosphorylation of full-length and deletion constructs of the constitutively active form of IKK2 harboring phosphomimetic Ser177Glu and Ser181Glu substitutions.

The online version of this article includes the following source data and figure supplement(s) for figure 1:

**Source data 1.** Original unedited autoradiograph and Coomassie-stained gel files used in *Figure 1C, D, E and F*.

**Source data 2.** Original autoradiographs and Coomassie-stained gel files used in *Figure 1C, D, E and F* with sample labels.

**Figure supplement 1.** Activity and purity level check of FL IKK2 WT and K44M.

**Figure supplement 1—source data 1.** Original unedited autoradiograph, Coomassie-stained gel, Silver-stained gel, western blot files and plots used in *Figure 1—figure supplement 1A, B, C and D*.

*Figure 1 continued on next page*

*Figure 1 continued*

**Figure supplement 1—source data 2.** Original autoradiograph, Coomassie-stained gel, Silver-stained gel, western blot files and plots used in *Figure 1—figure supplement 1A, B, C and D* with sample labels.

of signal-responsive S32/S36 of IκBα. The phosphorylation of IκBα at S32 was severely compromised upon substitution of Y169 of IKK2 to a phospho-ablative phenylalanine. Additionally, signal-responsive phosphorylation of IκBα upon TNF-α treatment in MEF cells reconstituted with IKK2-Y169F was severely diminished in comparison to wild type IKK2 (IKK2-WT). We also observed that an IKK2 bearing the K44M mutation resulted in a loss of S177 and S181 as well as tyrosine autophosphorylation activities and consequent loss of S32/S36 phosphorylation in IκBα, confirming a critical function of this lysine residue. Interestingly, phosphorylation of residues within the C-terminal PEST region of IκBα was retained in IKK2-K44M. Finally, we observed an intriguing activity of IKK2, when fully autophosphorylated at its serine and tyrosine residues, to phosphorylate IκBα in the absence of an exogenous supply of ATP, if ADP is present in the reaction. This result suggests the possibility of a unique, and likely transient, autophosphorylated form of IKK2 that can serve to relay phosphoryl group(s) specifically to S32/S36 of IκBα. Such a mechanism contrasts with the conventional transfer of γ-phosphate groups directly from ATP to the substrate observed in eukaryotic protein kinases.

## Results

### IKK2 undergoes autophosphorylation at uncharacterized sites

Signal-induced phosphorylation of activation loop residues Ser177 and Ser181 is the hallmark signature of IKK2 catalytic activity (*Figure 1B*). In contrast, hyperphosphorylation of other serine residues located within the flexible C-terminal region of IKK2, spanning amino acids 701–756 (*Figure 1B*), has been reported to down-regulate IKK2 activity in cells (*Delhase et al., 1999*). Ectopically over-expressed IKK2 has been observed to autophosphorylate the two serines of the activation loop in trans even in the absence of an activating physiological signal (*Polley et al., 2013*). We observed that purified recombinant full-length IKK2 is autophosphorylated when incubated with ATP in the absence of NEMO in a time-dependent manner, and likely at multiple sites as indicated by the diffuse nature of the slower migrating phospho-IKK2 band (*Figure 1C*). The autophosphorylated IKK2 produced a sharper band in the presence of NEMO – possibly reflecting enhanced specificity of phosphorylation (*Figure 1D*, left panel) – although NEMO did not alter the efficiency of phosphorylation for its *bona fide* substrate IκBα (*Figure 1D*, right panel). An in vitro kinase assay with a K44M mutant of IKK2 that lacks both its autophosphorylation and IκBα phosphorylation activities along with the native sequence IKK2 using γ-$^{32}$P-ATP confirmed that phosphorylation of IKK2 was self-catalyzed and not due to a spurious contaminating kinase (*Figure 1—figure supplement 1A*). K44 is an ATP-contacting residue that is conserved across STYKs, including IKK2, and its mutation to methionine typically reduces or eliminates kinase activity. An in vitro kinase assay showing inhibition of specific phosphorylation of S32/36 on IκBα and IKK2-autophosphorylation performed in the presence of an IKK-specific ATP-competitive inhibitor, Calbiochem Inhibitor VII, further validates that phosphorylation of IKK2 is due to its own activity (*Figure 1E*, *Figure 1—figure supplement 1B*). In these experiments, the reaction mixtures of kinase and kinase:substrate were incubated with the inhibitor for 30 min prior to the addition of the phosphate donor, γ-$^{32}$P-ATP or cold ATP. Purity of recombinant IKK2 and IKK2 K44M proteins is shown in *Figure 1—figure supplement 1C*, and an LC-MS/MS analysis of the IKK2 K44M is shown in *Figure 1—figure supplement 1D*.

Next, we performed kinase assays with variants of IKK2 protein with activation loop S177 and S181 mutated to phospho-mimetic glutamate resulting in the constitutively active IKK2-EE mutant form (*Zandi et al., 1998*) and with the C-terminal serine-rich region truncated to various extents. The shortest construct, Δ664EE, which lacked the entire flexible C-terminal region (670-756), was subjected to limited proteolysis using trypsin to further eliminate flexible N- and C-terminal residues and subsequently purified for in vitro kinase assays. These deletion IKK2-EE constructs displayed both autophosphorylation and substrate IκBα phosphorylation activities (*Figure 1F*). The observed autophosphorylation of IKK2 lacking serines of the activation loop and the C-terminal segment suggested the existence of hitherto uncharacterized sites of autophosphorylation in IKK2.

## Autophosphorylation of IKK2 reveals its dual specificity

We were intrigued as to whether the observed autophosphorylation activity of IKK2 might involve other residues in addition to the previously characterized serines, i.e., the possibility of IKK2 being a dual specificity kinase. Indeed, we observed by western blot with anti-phosphotyrosine antibodies the unexpected de novo autophosphorylation of IKK2 at tyrosine residues (*Figure 2A*, middle panel) in addition to the previously reported autophosphorylation of activation loop serines (*Figure 2A*, upper panel) when full-length native sequence IKK2 was incubated with $Mg^{2+}$-ATP for various amounts of time. Increased autophosphorylation of activation loop S177 and S181 over time reflects nonhomogeneous and incomplete phosphorylation of activation loop S177 and S181 in the IKK2 obtained from recombinant baculovirus-infected Sf9 insect cells. To address the possibility of non-specificity by the pan-phosphotyrosine antibody (used in *Figure 2A*), we further performed the experiment with an alternate anti-phosphotyrosine specific antibody that confirmed detection of de novo tyrosine phosphorylation was independent of the source of the antibody used (*Figure 2B*). ATP-dependent tyrosine autophosphorylation was not observed in the presence of an IKK-specific inhibitor, Inhibitor VII (*Figure 2C*), and with the kinase inactivating IKK2 K44M mutant (*Figure 2D*). Additionally, autophosphorylation assays performed in the presence of urea revealed tyrosine autophosphorylation to be undetectable at urea concentrations greater than 1 M (*Figure 2—figure supplement 1A*). More sensitive assays using γ-$^{32}$P-ATP recapitulated elimination of pan-autophosphorylation and IκBα substrate phosphorylation of IKK2 above 1 M urea (*Figure 2—figure supplement 1B*). These results demonstrate the dual specificity of recombinant IKK2 in phosphorylating serine residues on itself and on substrates, as well as tyrosine residues on itself. The fact that activities of IKK2 toward S32 and S36 of IκBα and its own tyrosine residue(s) were affected similarly by both general (AMPPNP and Staurosporine) and IKK-specific (MLN-120B, TPCA, Inhibitor VII; *Figure 2—figure supplement 1C*) kinase inhibitors indicates that both activities occur in the same active site. Using various truncated versions of IKK2, we further observed that the C-terminal NBD of IKK2 is not required for its autocatalytic dual specificity and in phosphorylating IκBα at S32 and S36 (*Figure 2E*) in vitro. Semi-quantitative assessment of phospho-Ser and phospho-Tyr residues on a constitutively active NBD-deficient IKK2 (IKK2Δ664EE) was performed by western blot with monoclonal anti-phosphoserine and anti-phosphotyrosine antibodies, which revealed de novo autophosphorylation at tyrosine residues but not at serines (*Figure 2F*). The phospho-ablative S177A/S181A (AA; non-phosphorylatable activation loop) mutant of IKK2, which is incapable of phosphorylating S32 and S36 of IκBα (*Figure 2—figure supplement 1D*), also lacked tyrosine autophosphorylation activity (*Figure 2G*) in an in vitro kinase assay, suggesting that phosphorylation of activation loop S177/S181 is critical for dual specificity of IKK2.

## Dual-specific autophosphorylation is critical for the function of IKK2

Previous reports suggested that phosphorylation of IKK2 on Y169, Y188, and/or Y199 is critical for its function (*Darwech et al., 2010*; *Meyer et al., 2013*). We performed mass spectrometric analyses on in vitro autophosphorylated IKK2 and identified Y169 to be the prominent tyrosine autophosphorylation site (*Figure 3—figure supplement 1A*). To explore the possible relevance of phosphorylation at tyrosine residues within the activation loop to kinase catalytic activity, we analyzed the experimentally determined X-ray crystal structures of IKK2 monomers within reported models of human IKK2 dimers (PDB ID: 4KIK, 4E3C) (*Liu et al., 2013*; *Polley et al., 2013*). In the active state conformers, the position and conformation of Tyr169 appears to be well positioned for accepting a phosphate from ATP intramolecularly (*Figure 3—figure supplement 1B*). A superposition of IKK2 with a pseudo-substrate-bound PKA shows that the hydroxyl of Tyr169 in IKK2 projects toward the γ-phosphate of ATP similarly to the serine hydroxyl on the PKA inhibitor pseudo-substrate (*Knighton et al., 1991*; *Nolen et al., 2004*; *Figure 3—figure supplement 1B and C*). Interestingly, the position occupied by Y169 in IKK2 primary sequence corresponds to the classic DFG +1 (DLG in cases of IKK2 and IKK1; *Figure 3A & B*). The DFG +1 position has been reported to be critical in defining the substrate specificity of a kinase (*Chen et al., 2014*).

We primarily focused our attention on the previously described K44M mutant and the newly generated Y169F mutant of IKK2, since these residues are located within the vicinity of the enzyme active site. We observed that the IKK2 K44M mutant fails to undergo autophosphorylation of tyrosine (*Figure 2D*) and the activation loop serines (*Figure 3—figure supplement 1D*). Surprisingly, IKK2 K44M displayed

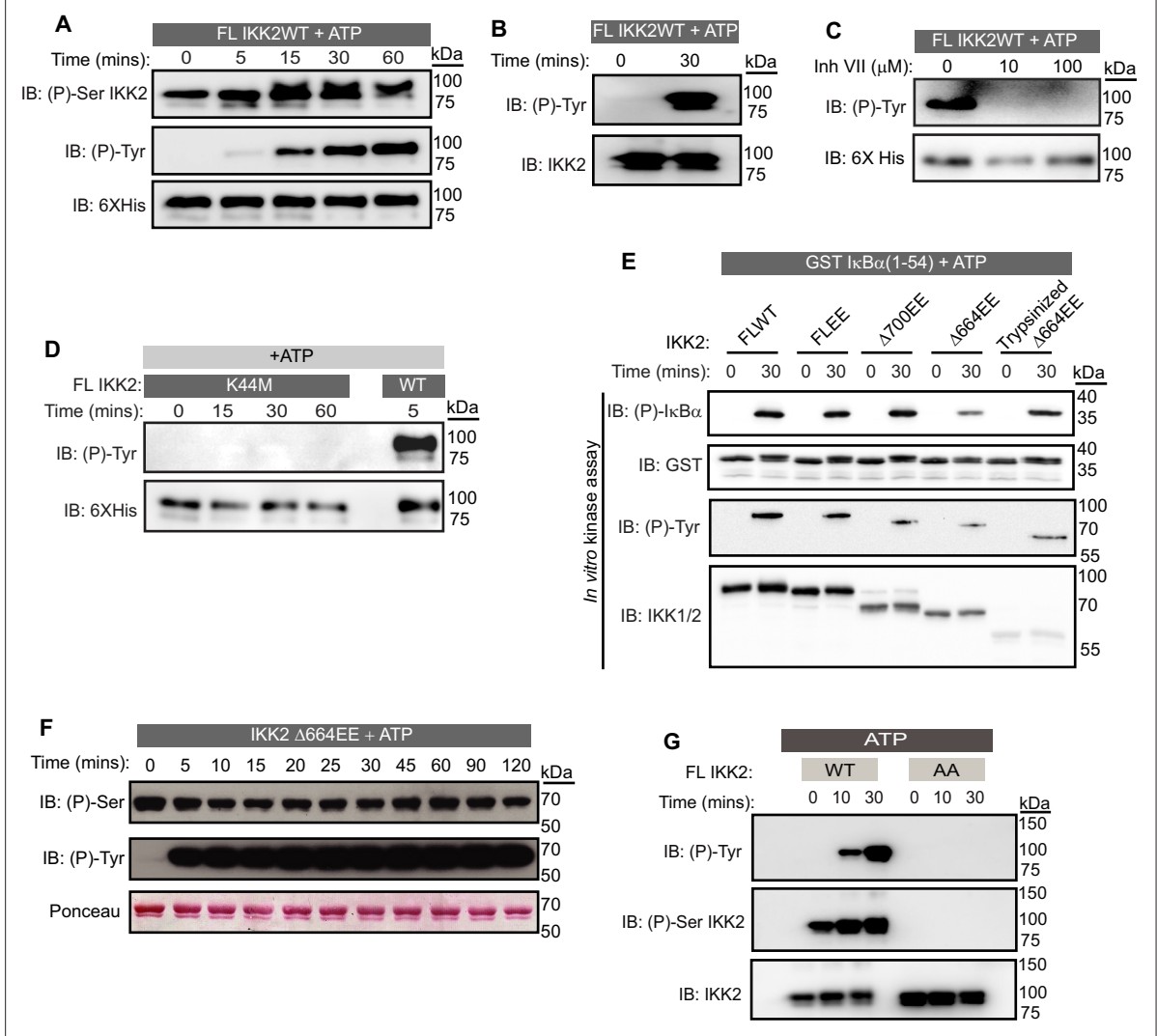

**Figure 2.** IKK2 displays autocatalytic dual specificity. (**A**) In vitro kinase assay with unlabeled ATP showing autophosphorylations in FL IKK2WT detected by immunoblotting using antibodies specific against phosphor-Ser (177/181) and phospho-Tyr residues. (**B**) pTyr on IKK2 detected using a different commercial source of phospho-Tyr antibody. (**C**) Effect of Inhibitor VII on tyrosine autophosphorylation of FL IKK2WT. (**D**) Autophosphorylation of IKK2 K44M mutant compared to that of IKK2 WT assessed at different time points through immunoblotting performed with phospho-Tyr antibody. (**E**) Autophosphorylation of tyrosines along with phosphorylation of GST-tagged IκBα (1-54) substrate with full-length and deletion mutants of IKK2 harboring phosphomimetic Ser177Glu and Ser181Glu substitutions. (**F**) In vitro de novo auto-phosphorylation of IKK2 Δ664EE construct on tyrosine analyzed by phospho-Ser and phospho-Tyr-specific monoclonal antibodies. (**G**) Autophosphorylations at tyrosine and AL-serine residues upon fresh ATP treatment of FLIKK2 WT and FLIKK2 S177A,S181A assessed by immunoblot analysis using antibodies against phospho-IKK2-Ser(177/181) and phospho-Tyr.

The online version of this article includes the following source data and figure supplement(s) for figure 2:

**Source data 1.** Original unedited western blot files used in *Figure 2A–G*.

**Source data 2.** Original western blot files used in *Figure 2A–G* with sample labels.

**Figure supplement 1.** Effect of Urea, kinase inhibitors and activation loop Serine mutations on IKK2 activity.

**Figure supplement 1—source data 1.** Original unedited autoradiograph, Coomassie-stained gel, and western blot files used in *Figure 2—figure supplement 1A–D*.

**Figure supplement 1—source data 2.** Original autoradiograph, Coomassie-stained gel, and western blot files used in *Figure 2—figure supplement 1A–D* with sample labels.

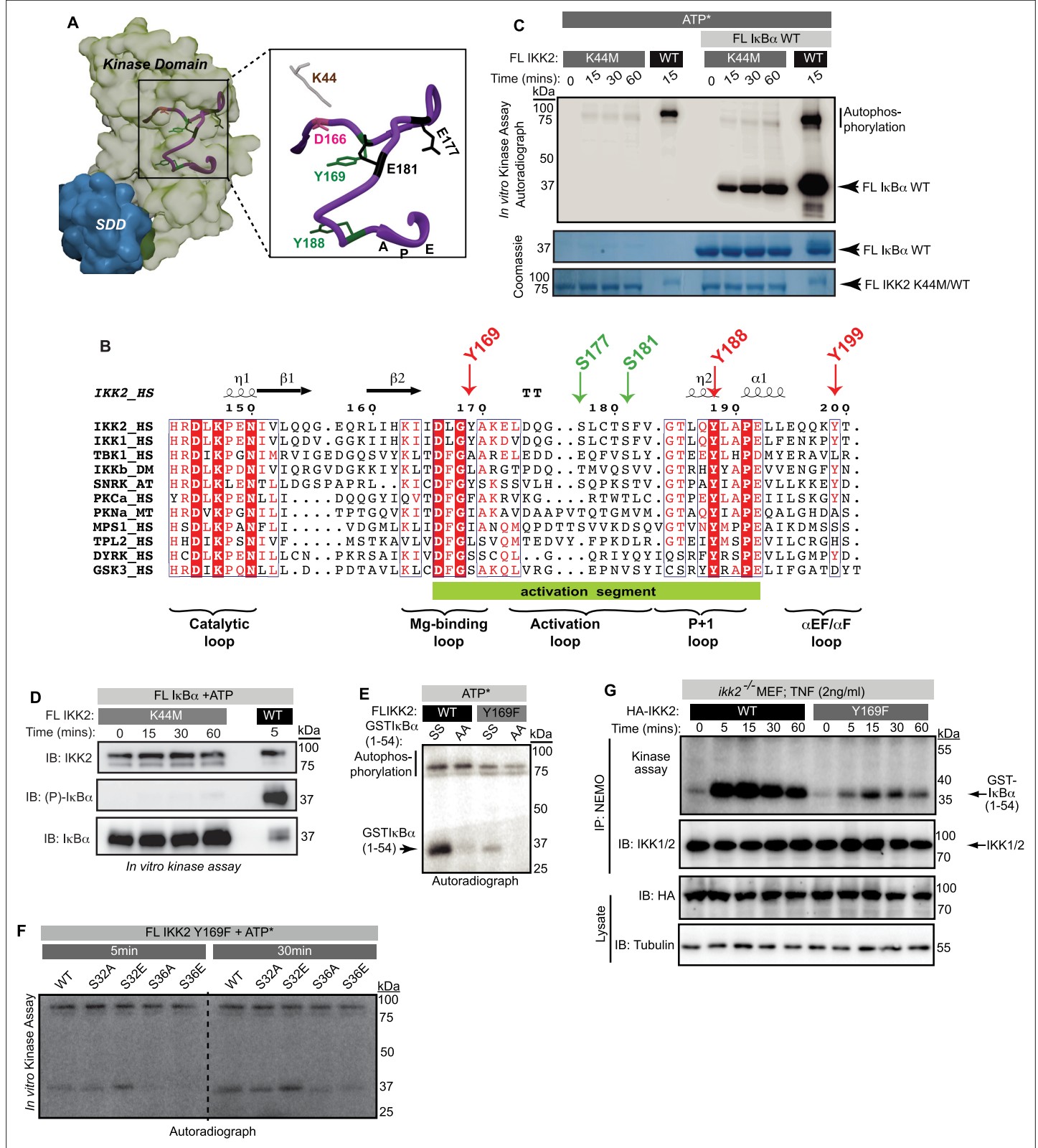

**Figure 3.** Dual-specific autophosphorylation is critical for the function of IKK2. (**A**) A surface representation of IKK2-KD structure (adapted from PDB ID 4E3C; KD is shown in light green and SDD in teal) with positions of canonically important residues within the AL (purple ribbon) marked. Tyrosine residues in the activation segment are marked in green, Tyr169 among which is identified to be autophosphorylated. (**B**) Amino-acid sequence alignment of activation loop segment of different kinases in the IKK-family. The tyrosine at position DFG +1 (DLG +1 in case of IKK1 and

*Figure 3 continued*

IKK2) is observed only in IKK and in the stress response-related plant kinase SnRK2, but not in other structural homologues of IKK or dual specificity kinases, for example DYRK and GSK3β (both contain Ser at that position). Tyr at position 188 (204 in PKA) is universally conserved. (**C**) Substrate and autophosphorylation activities of FL IKK2 WT and IKK2 K44M mutant were compared using in vitro radioactive kinase assay in presence and absence of FL IκBα WT as the substrate. (**D**) Specific residue-selectivity of phosphorylation by the FL IKK2 K44M analyzed using an antibody specific for phospho-S32/36 of IκBα. (**E**) In vitro kinase assay using radiolabeled ATP performed with IKK2 WT and IKK2 Y169F in the presence of WT and AA-mutant of GST-tagged IκBα (1-54) substrate. (**F**) In vitro kinase assay using radiolabeled ATP performed with IKK2 Y169F mutant in the presence of various GST-tagged IκBα(1-54) substrates indicating abolition of substrate phosphorylation in S36A and S36E mutants of IκBα. (**G**) Severe reduction of IKK activity with IKK immunoprecipitated (IP-ed) with anti-NEMO antibody from whole cell extract (n=2) of TNFα-induced *ikk2⁻/⁻* MEF-3T3 cells reconstituted with mutant Y169F IKK2 compared to the wild-type.

The online version of this article includes the following source data and figure supplement(s) for figure 3:

**Source data 1.** Original unedited autoradiograph, Coomassie-stained gel, and western blot files used in *Figure 3C, D, E, F, and G*.

**Source data 2.** Original autoradiograph, Coomassie-stained gel, and western blot files used in *Figure 3C, D, E, F, and G* with sample labels.

**Figure supplement 1.** Identification of Tyr169 phosphorylation, and effect of Y169F and K44M mutations on IKK2's specificity.

**Figure supplement 1—source data 1.** Original unedited autoradiograph, Coomassie-stained gel, and western blot files used in *Figure 3—figure supplement 1D–G*.

**Figure supplement 1—source data 2.** Original autoradiograph, Coomassie-stained gel, and western blot files used in *Figure 3—figure supplement 1D–G* with sample labels.

very weak (compared to that of WT IKK2) autophosphorylation activity but a more robust phosphorylation of full-length IκBα, albeit to a much lesser degree than native sequence IKK2, in a radioactive in vitro kinase assay (*Figure 3C*). Interestingly, we observed a lack of phosphorylation of IκBα at S32 and S36 by IKK2 K44M (*Figure 3D*). This suggests that IKK2 K44M phosphorylates IκBα at residues within its C-terminal PEST, consistent with the observed phosphorylation of IκBα (*Figure 3C & D*). Moreover, IKK2 K44M phosphorylated super-repressor IκBα S32A/S36A more efficiently than native sequence IκBα, in sharp contrast to IKK2 of native sequence (*Figure 3—figure supplement 1E*). This suggests that IKK2 K44M retains significant kinase activity toward IκBα, although it is incapable of specific IκBα phosphorylation at S32, S36. LC-MS/MS analysis with 20 µg of Sf9-derived IKK2 K44M protein used in our studies did not indicate the obvious presence of any contaminating kinase. It is noteworthy that an equivalent K to M mutant of Erk2 retained ~5% of its catalytic activity (*Robbins et al., 1993*).

Next, we assessed the activity of a tyrosine-phosphoablative Y169F mutant of IKK2 in an in vitro kinase assay and observed a severe reduction in both autophosphorylation and IκBα phosphorylation activities (*Figure 3E*). We measured its phosphorylation activity toward native sequence IκBα as well as IκBα with either S32 or S36 residues independently substituted to either phosphoablative alanine or phosphomimetic glutamic acid. IKK2 Y169F displayed drastically reduced levels of kinase activity toward S36A/E mutants of IκBα, while the S32A/E mutants were minimally affected when compared against the native IκBα substrate (*Figure 3F*). The reduction in phosphorylation of S36A/E by IKK2 was less pronounced than that by IKK2 Y169F (*Figure 3—figure supplement 1F*). These observations suggest that the tyrosine residue at position 169 (and possibly its phosphorylation) might contribute toward signal-responsive phosphorylation of IκBα.

We also measured signal-induced activation of IKK2 upon TNF-α treatment in *ikk2⁻/⁻* mouse embryonic fibroblast (MEF) cells reconstituted with native sequence IKK2 and IKK2 Y169F. Catalytic activity is significantly reduced in IKK2 Y169F (*Figure 3G*). Nonetheless, phosphorylation of activation loop serines was only marginally defective for IKK2 Y169F as judged by anti-phosphoserine-177 western blot (*Figure 3—figure supplement 1G*) hinting at the necessity, but not sufficiency, of activation loop phosphorylation for IKK2 to become fully active. Together, these results imply a possible interdependence of phosphorylation at Y169 of IKK2 and S32 of IκBα.

## Structural analyses of IKK2 autophosphorylation

We next analyzed three states of IKK2: unphosphorylated (UnP-IKK2), phosphorylated at S177 and S181 (p-IKK2), and phosphorylated at Y169, S177, and S181 (P-IKK2) using computational approaches of molecular dynamic (MD) simulations of 200 ns scale and flexible molecular docking (see methods). Phosphorylation at S177 and S181 increased folding stability of the kinase, which was further stabilized by phosphorylation at Y169 as evidenced by a gradual decrease in the total energy of the system (*Figure 4A*). Results of differential scanning calorimetry (DSC) in the presence of ADP vs. ATP also

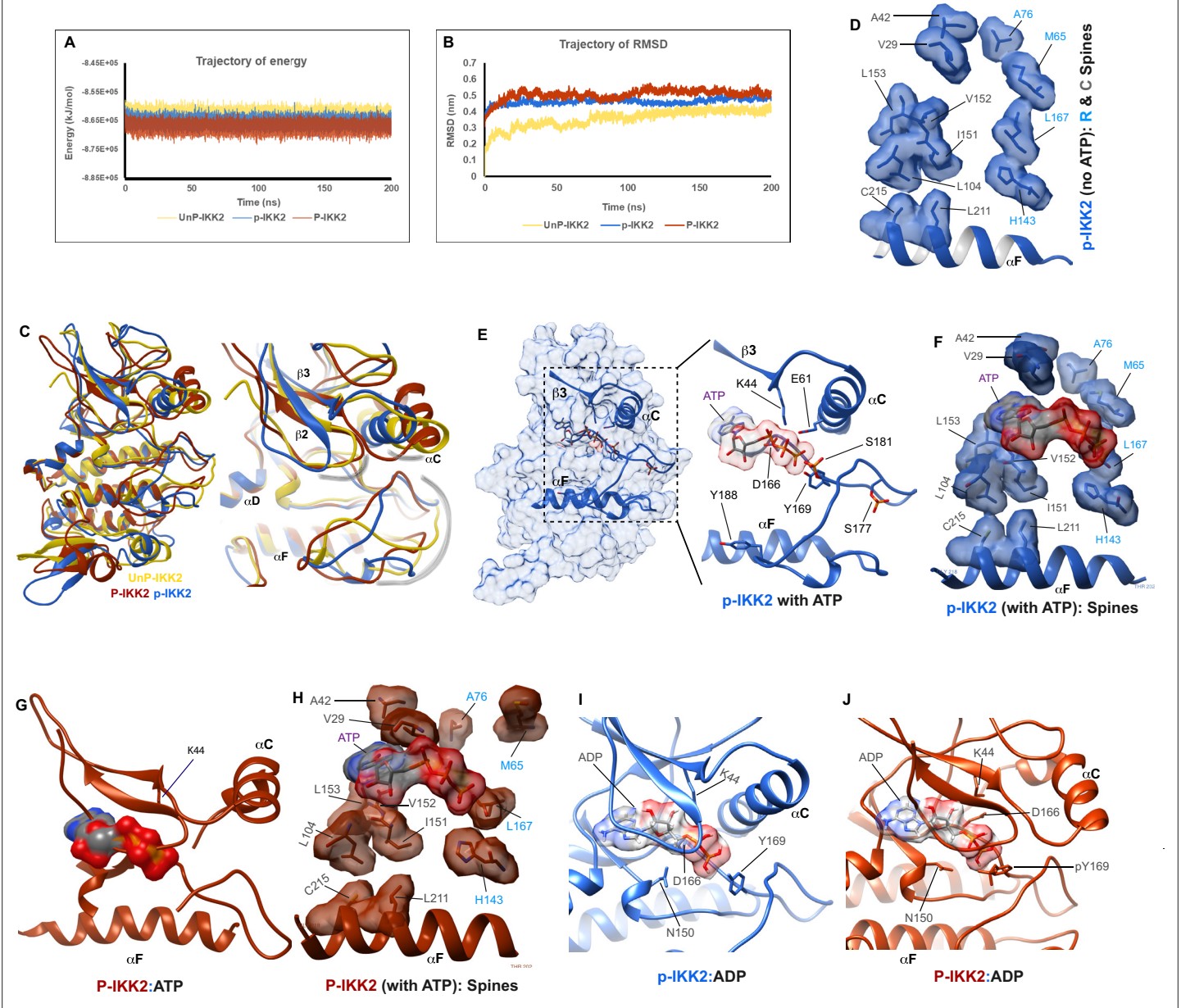

**Figure 4.** Structural analyses of IKK2 autophosphorylation. (**A**) Molecular dynamics (MD) simulations of three differently phosphorylated states of IKK2 - UnP-IKK2, p-IKK2, and P-IKK2, and 200ns trajectory of energy of these states shown in golden yellow, blue, and copper red, respectively. Same coloring scheme has been maintained in this figure and in the corresponding figure supplement (see *Supplementary file 3* for additional details). (**B**) 200 ns trajectory of RMSD of the three states (see *Supplementary file 4* for additional details). (**C**) Superposition of structures representing the differently phosphorylated states post 200 ns MD simulation. Relevant regions, for example activation loop, αC-helix, and Gly-rich loop areas are highlighted in the right panel. (**D**) Residues forming canonical R- (labeled in blue) and C-spines (labeled in dark gray) representing an active conformation in p-IKK2 are shown. (**E**) ATP-bound p-IKK2 structure is depicted. In the right panel, relevant residues are highlighted. The position of Tyr169 is conducive to autophosphorylation. (**F**) ATP maintains the desired continuum of R- and C-spines observed in active kinase conformations. (**G**) P-IKK2 cannot accommodate ATP in its cleft, and the αC-helix is displaced. (**H**) Met65 is moved away from the R-spine, failing to form canonically active conformation of R- and C-spines. (**I and J**) ADP-bound states of p-IKK2 and P-IKK2, both of which can accommodate ADP in their respective clefts.

The online version of this article includes the following source data and figure supplement(s) for figure 4:

**Source data 1.** Source PDB files used for creating *Figure 4C–J* and *Figure 4—figure supplement 1B and G*.

**Figure supplement 1.** Effect of IKK2 autophosphorylation on its thermal stability, dynamics and energetics of ATP/ADP binding.

indicated a change of Tm from ~40 °C to ~50 °C, reflecting a striking enhancement in stability upon autophosphorylation of IKK2 (*Figure 4—figure supplement 1A*). These results correlate with observations of increased solubility of IKK2 upon autophosphorylation; a property found to be essential for its successful crystallization (*Polley et al., 2013*).

Counter-intuitively, we observed the highest RMSD for P-IKK2, followed by p-IKK2, and then UnP-IKK2 (*Figure 4B*). This result might be indicative of increased internal motion despite phosphorylation-induced stabilization of IKK2 because of phosphorylation. The changes in total energy and RMSD values were consistent throughout the course of simulation, even though the magnitude was small. We observed distinct structural alteration within the KD upon phosphorylation at S177 and S181 or S177, S181, and Y169 (*Figure 4C*, models shown separately in *Figure 4—figure supplement 1B*). Analyses of the trajectory of RMSF values for the three states revealed different regions of the kinase domain with different RMSF values in different phosphorylated states (*Figure 4—figure supplement 1C*). We observed that, while the glycine-rich loop and the αC-helix in p-IKK2 were positioned in a manner reminiscent of an active kinase, those in UnP and P-IKK2 were oriented differently (*Figure 4C*). In fact, helix αC was found to be distorted in P-IKK2 in a manner that would make it unlikely to support canonical phosphotransfer from ATP. The activation loop in these three states adopted conformations distinct from each other. The activation loop in the UnP model was splayed out and moved closer to the tip of the C-lobe, whereas the activation loop in p-IKK2 and P-IKK2 moved inwards and adopted more compact conformations closer to the N-lobe (*Figure 4C*). The formation of the proper R- and C-spines in p-IKK2 confirmed its active conformation (*Figure 4D*). The p-IKK2 KD exhibited additional features consistent with an active kinase: a salt bridge between K44 and E61 (K72 and E91 in PKA, respectively), and the 'DFG in' conformation (although the sequence is DLG in IKK2). In addition, a dynamic cross-correlation matrix (DCCM) or contact map of each structure suggests that specific phosphorylation events render distinct allosteric changes to the kinase, even at locations distant from the phosphorylation sites (*Figure 4—figure supplement 1D*).

Next, we docked ATP onto IKK2 structures in all three states using LeDock and GOLD followed by rescoring with AutoDock Vina using the 0 ns (starting, S) and 200 ns (ending, E) structures of our MD simulation studies (see Materials and methods). The results indicated that ATP binding to P-IKK2 is relatively unfavorable (docking score in positive range) in both starting and ending structural models (*Figure 4—figure supplement 1E*), whereas ATP binding to UnP-IKK2 and p-IKK2 is favorable. ADP binding is, however, favorable for all phosphorylated or unphosphorylated IKK2 structural models except for the starting conformation of P-IKK2 (*Figure 4—figure supplement 1E*). We also extracted 50 intermediate structures from a 10 ns MD simulation run and calculated their binding free energies (ΔG) using the MM-PBSA (Molecular Mechanics Poisson-Boltzmann Surface Area) method. We again observed that ATP binding for the P-IKK2 population is highly unfavorable in contrast to their high relative preference for ADP, whereas p-IKK2 displayed comparable preference for both ATP and ADP (*Figure 4—figure supplement 1F*). In the ATP-docked structures, the ATP phosphate groups exhibited a pose (ΔG=−10.64 kcal/mol) very similar to that observed in the PKA structure (PDB ID: 1ATP) (*Figure 4—figure supplement 1G*), and the terminal phosphate was in close proximity to the Y169-OH, consistent with the likelihood of autophosphorylation at Y169 (*Figure 4E*). Further analyses of the ATP-bound p-IKK2 structure confirmed that the presence of ATP enhanced the formation of the R- and C-spines, as observed in active forms of kinases (*Figure 4F*). Superposition of the ATP from p-IKK2 on P-IKK2 indicated that narrowing of the canonical ATP-binding pocket in P-IKK2 may lead to rejection of ATP from the canonical binding pocket, triggered by severe clashes between the glycine-rich loop and ATP (*Figure 4G*). However, this does not exclude the possibility that P-IKK2 might interact with ATP through alternative binding modes. Furthermore, the R-spine in P-IKK2 exhibited a discontinuity (*Figure 4H*), with M65 (L95 in PKA) located away from the other three conserved spine residues and from the glycine-rich loop.

Binding of ATP to p-IKK2 could lead to autophosphorylation at Y169, thus generating P-IKK2 bound to ADP. The superposition of ADP (from the docked complex of p-IKK2 and ADP, ΔG=−9.88 kCal/mol, *Figure 4I*) onto the P-IKK2 structure confirmed that while P-IKK2 is unable to accommodate ATP in a manner similar to that observed in experimentally determined kinase structures, it could accommodate ADP into the ATP-binding cleft comfortably (*Figure 4J*). The affinities of ATP and ADP for p-IKK2 appear comparable as indicated by their respective binding energy/docking score values, like in other kinases (*Becher et al., 2013*). Thus, the cellular concentrations of ATP vs ADP could play

an important role in kinase activity. We speculate that ADP-bound P-IKK2, containing various and perhaps transiently phosphorylated groups within the activation loop and in close proximity to the active site, could act as an intermediate to relay phosphates back to ATP or to substrates.

These observations of IKK2 structural propensities hint at the possibility of residues within the flexible activation loop undergoing context-sensitive autophosphorylation and possibly rendering IKK2 with novel phospho-transfer activities distinct from the conventional direct transfer of γ-phosphate from ATP to substrate.

## Freshly autophosphorylated IKK2 phosphorylates IκBα even in the absence of ATP

We observed phosphorylation of tyrosine residue(s) only upon ATP treatment of purified IKK2 proteins, hinting at the likelihood of efficient cellular phosphotyrosine phosphatases (*Eckhart et al., 1979*). Surprisingly, crystal structures of Sf9-derived human IKK2 did not indicate phosphates on tyrosine residues even though recombinant IKK2 was treated with $Mg^{2+}$-ATP as part of the purification (PDB ID: 4E3C; *Polley et al., 2013*). It is unclear if the month-long period of crystallization at 18–20°C contributed to the loss of the generally very stable phosphotyrosine residues. This raises the intriguing possibility that phosphotyrosine residue(s) in IKK2 might somehow serve as a transient sink of phosphate for its eventual transfer to ADP or serines in the IκBα substrate. To investigate this hypothesis, we first incubated purified IKK2 with limiting amounts of radiolabeled γ-$^{32}$P-ATP for autophosphorylation and subsequently removed unreacted ATP via two passes through desalting spin columns (40 kDa MWCO). The nucleotide-free autophosphorylated IKK2 (P-IKK2) was then incubated with substrate IκBα either in the absence or with the fresh addition of unlabeled ATP/ADP. Resolution of the reaction mixture on an SDS-PAGE and subsequent autoradiography indicated transfer of radiolabeled phosphate to IκBα (*Figure 5A*), which was significantly enhanced in the presence of cold ADP (*Figure 5B*). Addition of excess cold ATP did not reduce the extent of IκBα phosphorylation, strongly suggesting the transfer of $^{32}$P-phosphate from $^{32}$P-IKK2 to IκBα (*Figure 5B*) and consistent with an IKK2-mediated phosphate relay mechanism. It is unclear if the transfer of phosphate from the phospho-IKK2 intermediate to IκBα occurs with or without direct involvement of ADP.

To further investigate the possibility of transfer of phosphate from the kinase (and not from ATP) to the substrate, we produced autophosphorylated IKK2 obtained by treatment of recombinant full-length IKK2 with excess cold ATP and purified using size-exclusion chromatography on a Superdex200 column to remove any traces of unreacted ATP (*Figure 5C*). P-IKK2 eluted at SEC fractions corresponding to MW between 670 and 158 kDa, much earlier than free ATP (*Figure 5D*). This P-IKK2 was incubated with IκBα substrate either in the presence or absence of ADP for varying time periods. A similar reaction was set up in the presence of ATP instead of ADP as a positive control. We observed, by western blot with sequence-specific anti-phosphoserine antibody, efficient phosphorylation of IκBα S32 and S36 residues in the presence of ADP (10 and 50 μM) but not in its absence. As anticipated, a robust phosphorylation of IκBα was observed in the presence of 50 μM ATP (*Figure 5E*). The very low-level detection of IκBα phosphorylation by radioactive assay in the absence of ADP and complete lack of signal in the cold immunoblotting assay (compare lanes marked with vertical arrows in *Figure 5B and E*) may indicate higher sensitivity of the radioactive assay. We quantified the intensities of each band in *Figure 5E*, and the ratios of the normalized average intensities of the tyrosine phosphorylated IKK2 and S32/36-phosphorylated IκBα with their respective full-length protein controls were plotted as shown in *Figure 5—figure supplement 1A*. Next, we followed the same experimental scheme as shown in *Figure 5C* and assessed IκBα phosphorylation by relay mechanism at a finer time interval. Phosphorylation of S32/S36 of IκBα increased with time as shown in *Figure 5—figure supplement 1B*. The requirement for ADP in the observed phospho-transfer prompted us to consider if our ADP was contaminated with trace amounts of ATP. ESI-MS analysis of our 50 μM ADP solution did not detect any peak corresponding to ATP (*Figure 5—figure supplement 1C*). Dependence of IκBα phosphorylation on ADP raises the possibility of infinitesimal reversibility generating ATP from ADP in the active site of the kinase. We could not detect the generation/presence of ATP in a setting similar to that described in *Figure 5A & B* by TLC-based analysis (*Figure 5—figure supplement 1D*).

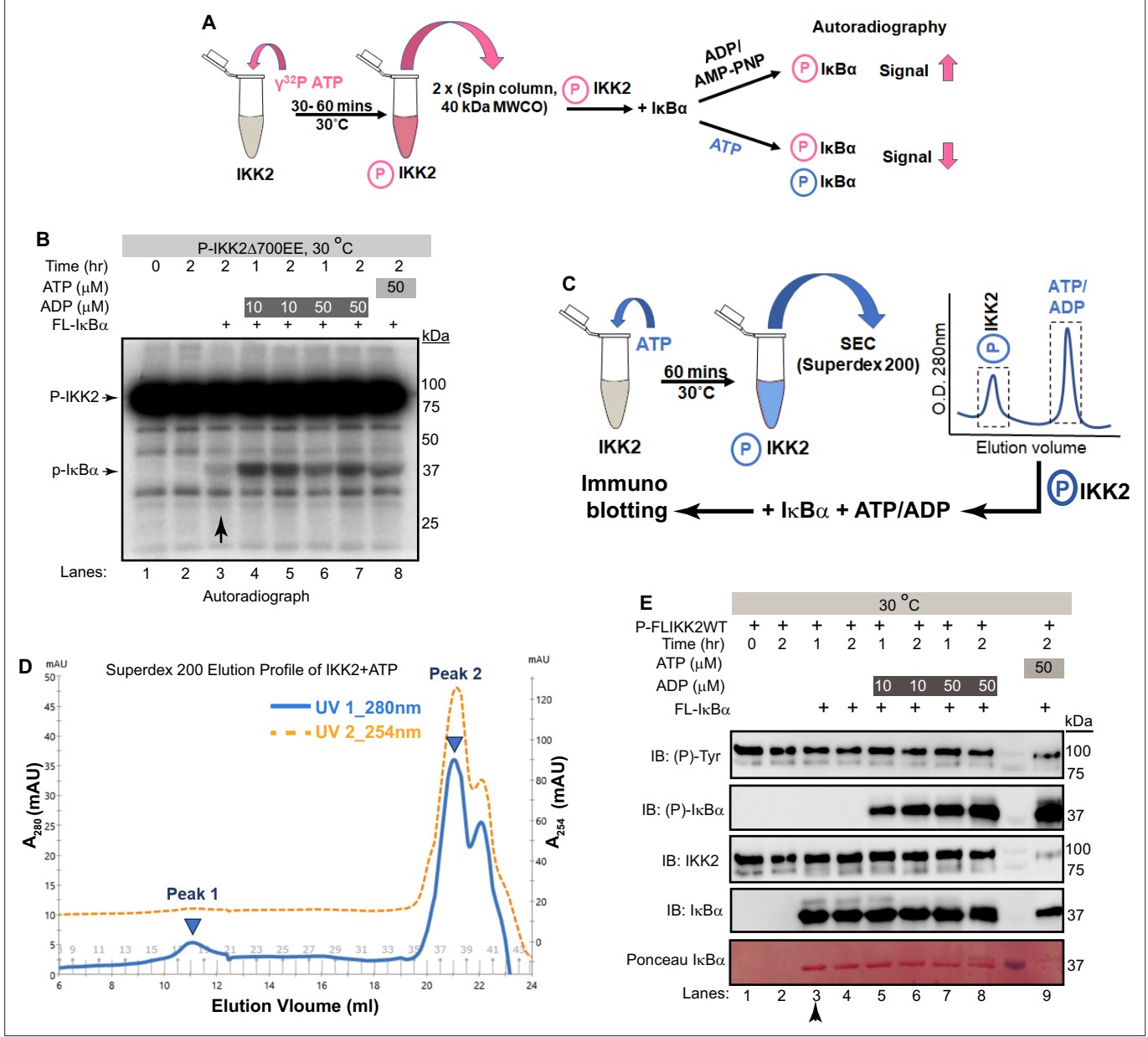

**Figure 5.** Freshly autophosphorylated IKK2 relays phosphates to IκBα. (**A**) A schematic of the autoradiography experiment to monitor the path of phosphate(s) from phospho-IKK2 to substrate. (**B**) Autophosphorylated (with radiolabeled ATP) purified IKK2 could transfer its phosphate to IκBα substrate in the absence of any nucleotide, and the transfer efficiency is enhanced upon addition of ADP or ATP. (**C**) A schematic of the immunoblot experiment to monitor the path of phosphate(s) from phospho-IKK2 to substrate. (**D**) Elution profile in size-exclusion chromatography (Superdex200 10/30 increase) of phospho-IKK2 (Peak 1) to remove excess unlabeled ATP (Peak 2). Phospho-IKK2 from Peak1 was used in downstream phosphotransfer assays. (**E**) Immunoblotting experiment using specific antibodies indicated in the figure (n=2) showing that purified autophosphorylated (with cold ATP) IKK2 transfers its phosphate to IκBα substrate, and this transfer efficiency is enhanced upon addition of ADP or ATP.

The online version of this article includes the following source data and figure supplement(s) for figure 5:

**Source data 1.** Original unedited autoradiograph and western blot files used in *Figure 5B and E*.

**Source data 2.** Original autoradiograph and western blot files used in *Figure 5B and E* with sample labels.

**Figure supplement 1.** Phosphotransfer from autophosphorylated IKK2 to IκBα does not involve ATP-regeneration.

**Figure supplement 1—source data 1.** Original unedited western blot and TLC files used in *Figure 5—figure supplement 1B and D*.

**Figure supplement 1—source data 2.** Original western blot and TLC files used in *Figure 5—figure supplement 1B and D* with sample labels.

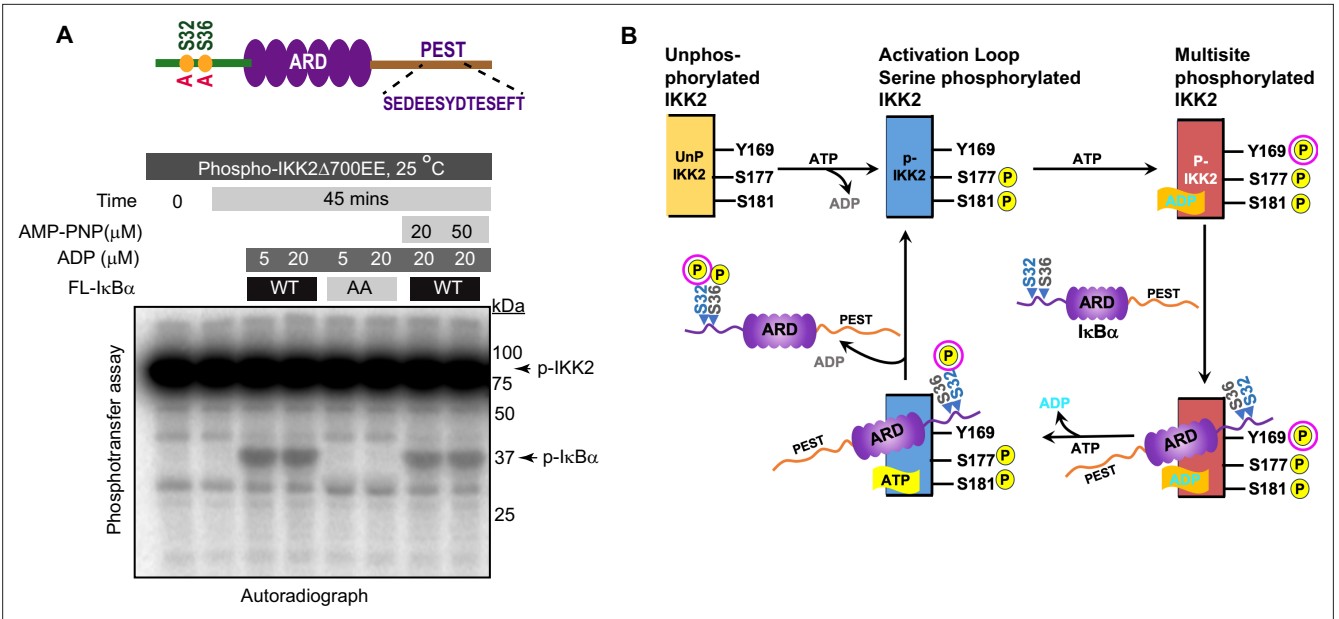

**Figure 6.** Specificity and fidelity of phosphotransfer by IKK2 to IκBα. (**A**) Autoradiograph showing phosphotransfer to full-length IκBα WT but not to its S32A,S36A double mutant. Domain organization and position of relevant S/T/Y residues of IκBα are shown above the autoradiograph. Also, AMP-PNP can support efficient phosphotransfer. (**B**) A proposed scheme of reactions during signal-responsive IKK2-activation and subsequent specific phosphorylation of IκBα at S32, S36 through phosphotransfer in the presence of ADP.

The online version of this article includes the following source data and figure supplement(s) for figure 6:

**Source data 1.** Original unedited autoradiograph file used displayed in *Figure 6A*.

**Source data 2.** Original autoradiograph file used displayed in *Figure 6A* with sample labels.

**Figure supplement 1.** Purified autophosphorylated (radiolabeled) FL IKK2 Y169F fails to transfer phosphate to IκBα substrate in absence or presence of ADP (n=1).

**Figure supplement 1—source data 1.** Original unedited autoradiograph file used in *Figure 6—figure supplement 1*.

**Figure supplement 1—source data 2.** Original autoradiograph file used in *Figure 6—figure supplement 1* with sample labels.

## Phosphotransfer from P-IKK2 is critical to the fidelity and specificity of IκBα phosphorylation

We next tested whether the observed phosphotransfer to IκBα is restricted only to the S32 and S36 residues required for NF-κB activation or also directed toward serine/threonine residues within the C-terminal PEST domain of IκBα. Kinase assays performed with P-IKK2 and IκBα of native sequence or harboring S32A and S36A mutations (AA) in the presence of γ-$^{32}$P-ATP revealed a lack of phosphorylation of IκBα AA, although that protein retains the serines and threonines of the C-terminal PEST region (*Figure 6A*). Furthermore, a similar assay with IKK2-Y169F showed its inability to transfer phosphate to IκBα (*Figure 6—figure supplement 1*). Taken together, this hints at the transfer of phosphate exclusively to S32 and S36 of IκBα from Y169. We also observed no inhibitory effect on the phosphotransfer by AMPPNP, an unhydrolyzable analogue of ATP, irrespective of the presence of ADP. This suggests that diverse adenine nucleotides prompt phosphotransfer by the kinase (*Figure 6A*). It is noteworthy that a recent study reported that IKK2 phosphorylates IκBα sequentially at S32 followed by S36 following a single binding event and that phosphorylation of S32 increases the phosphorylation rate of S36 (*Stephenson et al., 2023*).

Taken together with our results, we propose that catalytically active IKK2, that is by definition phosphorylated at S177 and S181, can undergo autophosphorylation on tyrosine residues within its activation loop. Furthermore, the tyrosine autophosphorylated IKK2 could serve as a phosphoenzyme intermediate to transfer its phosphate group to S32 of IκBα, thereby enhancing the rate of phosphorylation at S36, possibly using ATP for the second site phosphorylation at S36 (*Figure 6B*).

## Discussion

The phosphorylation of cognate substrates by protein kinases is regulated in a variety of ways (*Cullati et al., 2022*; *Sang et al., 2022*). IKK2, a kinase central to inflammation, phosphorylates serine residues of its primary substrate IκBα using γ-phosphate of ATP. In this study, we report several intriguing structural and biochemical properties of IKK2 that contribute to its substrate phosphorylation specificity and efficiency: (1) dual-specificity of IKK2 that entails autophosphorylation of its own tyrosine residues, in addition to known autophosphorylation of activation loop serines; (2) loss of specificity, but retention of diminished catalytic activity, upon disruption of a universally conserved salt bridge mediated by K44 in IKK2; and (3) relay of phosphate to substrate IκBα from autophosphorylated IKK2 (P-IKK2).

### Dual specificity of IKK2

Our present analyses of IKK2 revealed a surprising property wherein IKK2 phosphorylated at activation loop serines S177/S181 could be further autophosphorylated by ATP at multiple sites, including at least one of its tyrosine residues, yielding a hyper-phosphorylated form (P-IKK2) (*Figure 6B*). This defines IKK2 as an autocatalytic dual-specificity kinase rather than simply a prototypical Ser/Thr kinase. Several members of the Ser/Thr kinase family have been reported to undergo tyrosine phosphorylation, and it appears that many of these kinases employ diverse strategies for phosphorylation (*Bhattacharyya et al., 2006*; *Ge et al., 2002*; *Lochhead, 2009*; *Lochhead et al., 2006*; *Lochhead et al., 2005*; *Sugiyama et al., 2019*; *Tigno-Aranjuez et al., 2010*). Tyrosine residues within the activation loop of IKK2 are also reported to undergo signal-induced phosphorylation in cells, and this is proposed to be mediated by a tyrosine kinase (*Meyer et al., 2013*; *Otero et al., 2008*; *Rieke et al., 2011*). However, dual specificity is notably manifested by DYRK and GSK3β, even though the function of tyrosine phosphorylation in DYRK/GSK3β does not appear to be related to that of IKK2.

### The phosphorelay mechanism

The Ser/Thr/Tyr family of eukaryotic protein kinases transfer γ-phosphate of ATP directly to substrates, usually at S, T, and Y residues (*Cohen, 2002*; *Fischer and Krebs, 1955*; *Hunter, 1991*; *Krebs and Fischer, 1955*; *Nolen et al., 2004*). In contrast, histidine kinases found in prokaryotes and in some simple eukaryotes employ a mechanism in which the γ-phosphate of ATP is transferred first to an aspartate residue on a response regulator (RR) substrate through an internal phosphohistidine intermediate (*Borkovich and Simon, 1990*; *Gao and Stock, 2009*; *Hunter, 2022*; *Kalagiri and Hunter, 2021*; *Laub et al., 2007*; *Robbins et al., 1993*). Surprisingly, the hyper-phosphorylated P-IKK2 appears to display an ability to transfer phosphate(s) to substrate IκBα in the presence of ADP without requiring an exogenous supply of fresh ATP – hinting at the possibility of IKK2 acting as a phosphate sink capable of relaying phosphate from the phosphoenzyme intermediate to the substrate (*Figure 5B and E*). Since this phosphotransfer to S32 and S36 of IκBα is observed with autophosphorylated, constitutively active IKK2 EE, the transferred phosphates are unlikely to derive from phosphorylated S177 or S181. Phosphorylated Y169, on the other hand, could be a suitable candidate to act as an intermediate and a source of transferable phosphate due to its conducive location at the DFG +1 (DLG in IKK2) position. Y169 seems particularly fit for this role as it is the only active site-proximal residue prominently autophosphorylated within the activation loop of IKK2 (Figure S3A). Tyrosine at this position is unique only to IKK1 and IKK2 and the plant kinase SNRK1 and is absent even in close mammalian orthologs such as IKKε and TBK1 (*Larabi et al., 2013*; *Tu et al., 2013*). Interestingly, a study has implicated the role of DFG +1 residue in differentiating kinase specificity toward Ser versus Thr residues (*Chen et al., 2014*). These observations hint at the possibility of Y169 being involved in phosphorylation selectivity of IκBα substrate. We observed a role for Y169 in phosphorylating S32 of IκBα using both in vitro and ex vivo experiments (*Figure 3F & G*); however, our data does not confirm if Y169 phosphorylation is the sole contributor to specific signal-responsive phosphorylation of IκBα. It is noteworthy that an S32I mutant of IκBα containing the phosphoablative mutation at S32 is associated with ectodermal dysplasia and T-cell immunodeficiency (*Courtois et al., 2003*; *Mooster et al., 2015*). Autophosphorylation in the kinase domain has been observed to regulate function in other kinases as well, for example autophosphorylation at T220 greatly influenced both the activity and substrate specificity of CK1 (*Cullati et al., 2022*). So far, IKK2 appears to be the only other metazoan protein kinase that derives its substrate specificity through the formation of a phosphoenzyme intermediate apart from MHCK of *Dictyostelium discoideum* (an atypical eukaryotic protein kinase; *Ye et al., 2010*). The transfer of phosphate

from Tyr169 to Ser32 of IκBα represents an intriguing and unprecedented mechanism among metazoan protein kinases. We propose a possible explanation for the process: the hydroxyl group of Tyr169 is optimally positioned to readily accept the γ-phosphate from ATP at the expense of minimal energy in the presence or absence of the substrate IκBα. This relay of phosphate – from ATP to Tyr169 to Ser32 – might offer a more energetically favorable pathway than a direct transfer of phosphate from ATP to the substrate. Another intriguing feature of phosphorelay in IKK2 is its dependence on ADP (*Figure 5B & E*), which the generic kinase inhibitor AMPPNP (unhydrolyzable ATP analogue) failed to inhibit (*Figure 6A*). Taken together, we surmise that ADP or AMPPNP might act as an ATP surrogate in helping IKK2 adopt a conformational state commensurate with selective phosphorylation of S32 and S36 in IκBα. A similar phenomenon is reported for IRE1, where ADP was found to be a potent stimulator of its ribonuclease activity, and AMPPNP was also reported as a somewhat effective potentiator (*Lee et al., 2008*; *Sidrauski and Walter, 1997*). We speculate that Ser32 cannot be efficiently phosphorylated by p-IKK2:ATP due to a deliberate chemical mismatch of the hydroxyl group of Ser32 within the active site. However, it aligns well with the P-IKK2:ADP intermediate. Indirect supporting evidence, as described above, underscores the existence of a unique phosphotransfer mechanism; however, detailed structural, biochemical, and MD simulations studies are necessary to validate this model.

## Signaling specificity and digital activation

The amplitude, duration, and kinetics of IKK activity correlate with the strength of input signal, level of IκBα degradation, and NF-κB activation (*Behar and Hoffmann, 2013*; *Cheong et al., 2006*). This digital (all-or-none) activation profile of NF-κB appears to be due to rapid activation and inactivation of IKK2. The primary target of IKK2 is IκBα, which underlies the rapid activation of NF-κB, although multiple other substrates are targeted in vivo. This digital activation profile may be intrinsically linked to the phosphorelay process. The selection of specific IκBα residues for phosphorylation is unclear, although NEMO is reported to have a direct role (*Schröfelbauer et al., 2012*). We observe that P-IKK2 transfers phosphate(s) efficiently and specifically to S32 and S36 of IκBα, but not to other phosphorylatable sites, such as the serine and threonine residues within the IκBα C-terminal PEST region (*Figures 5B, E and 6A*). We surmise that the transiency of various phosphorylated residues in P-IKK2 could underlie the activation-inactivation event during signal response. In this context, it will be worth investigating if functional regulation of IKK2 involves properties analogous to that of the two-component histidine kinase-effector systems (*Bhate et al., 2015*; *Lamarche et al., 2008*; *Laub and Goulian, 2007*). It is noteworthy that phosphate transfer directly from phosphohistidine is energetically far more favorable than from phosphotyrosine, except in enzymatic processes. The phosphotransfer from P-IKK2 to IκBα investigated and reported herein reflects a single turnover event in the absence of the phosphate donor, ATP (*Figure 5B*). Not every phosphorylated residue over the long span of P-IKK2 is expected to have the ability to transfer its phosphate to IκBα and additionally, the P-IKK2 likely represents a pool with a non-homogeneous distribution of phosphorylated residues. This could be a reason for the observance of the sub-stoichiometric phospho-relay.

## The K44-conserved salt bridge is a determinant of specificity but not activity

We also observe a critical role of K44 in autophosphorylation of IKK2 and specific phosphorylation of S32 and S36 in IκBα. Interestingly, the IKK2 K44M mutant retained its non-specific kinase activity toward the C-terminal PEST region of IκBα. Along these lines, an engineered monomeric version of IKK2 (lacking its NBD and major portions of its SDD) was observed to retain specificity for the N-terminal serines of IκBα (*Hauenstein et al., 2014*), whereas a smaller version of IKK2 with only the KD and ULD lacked S32 and S36 specificity but retained the ability to phosphorylate IκBα within its C-terminal PEST region (*Shaul et al., 2008*). It is possible that a particular structural state of IKK2 or its interaction with a partner protein could direct IKK2 to phosphorylate specific sites within different substrates.

## Conclusion

It is intriguing how the activation of NF-κB, through unique phosphorylation events catalyzed by IKK2, appears to be regulated by a complex and multi-layered fail-safe mechanism. In the absence of

upstream signals, the IKK complex is unable to phosphorylate Ser 32/36 of IκBα efficiently, thereby keeping a check on the aberrant and untimely activation of NF-κB. Upon encountering proinflammatory cues, cells need to activate NF-κB immediately, efficiently, and specifically, that is IKK needs to be activated and phosphorylate Ser 32/36 of IκBα. NEMO warrants IKK activation and ensures that the IKK complex specifically chooses IκBα from a large pool of substrates of IKK2. IKK2 is designed in such a manner that it phosphorylates itself at an activation loop tyrosine when activated, such that phosphate group(s) can be relayed directly to Ser32/36 of IκBα with great fidelity, thus leaving little chance of a misconstrued signaling event thereby confirming NF-κB activation. Our discovery of an intriguing phosphotransfer reaction that helps accomplish the desired specificity could present the beginning of a new aspect in eukaryotic cell signaling by EPKs.

## Materials and methods

**Key resources table**

| Reagent type (species) or resource | Designation | Source or reference | Identifiers | Additional information |
|---|---|---|---|---|
| Gene (*Homo sapiens*) | IKBKB | GenBank | Gene ID: 3551 | |
| Gene (*Homo sapiens*) | NFKBIA | GenBank | Gene ID: 4792 | |
| Gene (*Homo sapiens*) | IKBKG | GenBank | Gene ID: 8517 | |
| Strain, strain background (*Escherichia coli*) | Rosetta2 (DE3) | Other | | Chemically competent cells |
| Strain, strain background (*Escherichia coli*) | DH5α | New England Biolabs | Cat# C2987H | Chemically competent cells |
| Strain, strain background (*Spodoptera frugiperda*) | Sf9 cells | Thermo Fischer | Cat# 12659017 | |
| Antibody | Anti-IκBα (L35A5) (Mouse monoclonal) | Cell Signaling Technology | Cat# 4814, RRID:AB_390781 | WB (1:2000) |
| Antibody | Anti-Phospho-IκBα (Ser$^{32/36}$) (5A5) (Mouse monoclonal) | Cell Signaling Technology | Cat# 9246, RRID:AB_2267145 | WB (1:2500) |
| Antibody | Anti-Phospho-IKKα/β (Ser$^{176/180}$) (16A6) (Rabbit monoclonal) | Cell Signaling Technology | Cat# 2697, RRID:AB_2079382 | WB (1:2000) |
| Antibody | Anti-Phospho-Tyrosine (Mouse monoclonal) | BD Biosciences | Cat# 610000, RRID:AB_397423 | WB (1:500) |
| Antibody | Anti-Phospho-Tyrosine (Mouse monoclonal) | EMD Millipore | Cat# 05–321 X | WB (1:1000) |
| Antibody | Anti-IKKβ (Rabbit Polyclonal) | Bio Bharti Life Science | Cat# BB-AB0094 | WB (1:2000) |
| Antibody | Anti-IKKβ (10AG2) (Mouse monoclonal) | Novus Biologicals | Cat# NB100-56509 | WB (1:5000) |

*Continued on next page*

*Continued*

| Reagent type (species) or resource | Designation | Source or reference | Identifiers | Additional information |
|---|---|---|---|---|
| Antibody | Anti-6XHis (Rabbit Polyclonal) | Bio Bharti Life Science | Cat# BB-AB0010 | WB (1:2500) |
| Antibody | Anti-GST (Rabbit polyclonal) | BioLegend | Cat# 924801, RRID:AB_2565461 | WB (1:10000) |
| Antibody | Anti-Phospho-Serine Q5 (Mouse Monoclonal) | Qiagen | Cat# 37430 | WB (1:2000) |
| Antibody | Anti-NEMO (Mouse monoclonal) | BD biosciences | Cat# 559675, RRID:AB_397297 | IP (1:1AB_397297AB_397297AB_397297AB_397297AB_39729700) |
| Antibody | Anti-HA 11 epitope (Mouse monoclonal) | BioLegend | Cat# 901502, RRID:AB_2565007 | WB (1:2000) IP (1:100) |
| Antibody | Anti-Tubulinβ3 (Mouse monoclonal) | BioLegend | Cat# 657402, RRID:AB_2562570 | WB (1:10,000) |
| Recombinant DNA reagent | pFastBac HT B (plasmid) | Invitrogen | | |
| Recombinant DNA reagent | pET24d (6xHis-TEV) (plasmid) (modified) | Other | | Prepared in lab |
| Sequence-based reagent | IKK2_1_F | This paper | PCR primer | 5′-GCATGATCAAGCTG GTCACCTTCCCTGAC-3′ |
| Sequence-based reagent | IKK2_700_R | This paper | PCR primer | 5′-ATAAGAATGCGGCCG CTCACTCAGGTAAGCTGTTGGAGGCCG-3′ |
| Sequence-based reagent | IKK2_AA_F | This paper | PCR primer | 5′-GCCAAGGAGCTGGAT CAGGGCGCTCTTTGCACAGCATTCGTGGGGACCCTGCAGTAC-3′ |
| Sequence-based reagent | IKK2_AA_R | This paper | PCR primer | 5′- GTACTGCAGGGTCCC CACGAATGCTGTGCAAAGAGCGCCCTGATCCAGCTCCTTGGC-3′ |
| Sequence-based reagent | IKK2_Y169F_F | This paper | PCR primer | 5′-CACAAAATTATTGACC TAGGATTTGCCAAGGAGCTGGATCAGGGC-3′ |
| Sequence-based reagent | IKK2_Y169F_R | This paper | PCR primer | 5′-GCCCTGATCCAGCTC CTTGGCAAATCCTAGGTCAATAATTTTGTG-3′ |
| Sequence-based reagent | NEMO_F | This paper | PCR primer | 5′-CGGAATTCATG AATAGGCACCTCTGG-3′ |
| Sequence-based reagent | NEMO_R | This paper | PCR primer | 5′-ACGCGTCGACTCA CTCAATGCACTCCATG-3′ |
| Sequence-based reagent | IκBα_F | This paper | PCR primer | 5′-CGCGGATCCATGTT CCAGGCGGCCGAG-3′ |
| Sequence-based reagent | IκBα_R | This paper | PCR primer | 5′-ACGCGTCGACTCAT AACGTCAGACGCTG-3′ |
| Peptide, recombinant protein | FL IKK2 WT | (*Polley et al., 2013*) PMID:23776406 | | Sf9-baculovirus expression system |
| Peptide, recombinant protein | FL IKK2 K44M | This paper | | Sf9-baculovirus expression system |

*Continued on next page*

*Continued*

| Reagent type (species) or resource | Designation | Source or reference | Identifiers | Additional information |
|---|---|---|---|---|
| Peptide, recombinant protein | FL IKK2 AA | This paper | | Sf9-baculovirus expression system |
| Peptide, recombinant protein | FL IKK2 Y169F | This paper | | Sf9-baculovirus expression system |
| Peptide, recombinant protein | FL IKK2 EE | (*Polley et al., 2013*) PMID:23776406 | | Sf9-baculovirus expression system |
| Peptide, recombinant protein | (1-700) IKK2 EE | (*Polley et al., 2013*) PMID:23776406 | | Sf9-baculovirus expression system |
| Peptide, recombinant protein | (1-664) IKK2 EE | (*Shaul et al., 2008*) PMID:18657515 | | Sf9-baculovirus expression system |
| Peptide, recombinant protein | FL IκBα WT | (*Hauenstein et al., 2014*) PMID:24611898 | | Bacterial expression system |
| Peptide, recombinant protein | FL IκBα AA | (*Hauenstein et al., 2014*) PMID:24611898 | | Bacterial expression system |
| Peptide, recombinant protein | (1-54) IκBα WT | (*Polley et al., 2013*) PMID:23776406 | | Bacterial expression system |
| Peptide, recombinant protein | (1-54) IκBα S32A | This paper | | Bacterial expression system |
| Peptide, recombinant protein | (1-54) IκBα S32E | This paper | | Bacterial expression system |
| Peptide, recombinant protein | (1-54) IκBα S36A | This paper | | Bacterial expression system |
| Peptide, recombinant protein | (1-54) IκBα S36E | This paper | | Bacterial expression system |
| Peptide, recombinant protein | (1-54) IκBα AA | (*Hauenstein et al., 2014*) PMID:24611898 | | Bacterial expression system |
| Peptide, recombinant protein | FL NEMO WT | This paper | | Bacterial expression system |
| Commercial assay or kit | PureLink HiPure Plasmid Miniprep Kit | Invitrogen | Cat# K210002 | For Bacmid prep |
| Commercial assay or kit | QIAquick Gel Extraction Kit | Qiagen | Cat# 28704 | For Gel extraction |
| Chemical compound, drug | IKK Inhibitor VII | Calbiochem | CAS 873225-46-8 | IKK inhibitor |
| Chemical compound, drug | Staurosporine | Sigma-Aldrich | S5921 | Kinase inhibitor |

*Continued on next page*

*Continued*

| Reagent type (species) or resource | Designation | Source or reference | Identifiers | Additional information |
|---|---|---|---|---|
| Chemical compound, drug | TPCA | Abcam | ab145522 | IKK inhibitor |
| Software, algorithm | GROMACS | GROMACS | RRID:SCR_014565 | |
| Software, algorithm | UCSF Chimera | UCSF Chimera | RRID:SCR_004097 | |
| Software, algorithm | AutoDock Vina | AutoDock Vina | RRID:SCR_011958 | |
| Software, algorithm | CHARMM | CHARMM | RRID:SCR_014892 | |

## Reagents and cell culture materials

Monoclonal mouse anti-IκBα (L35A5) (catalogue number: 4814, RRID:AB_390781), mouse anti-Phospho-IκBα (Ser$^{32/36}$) (5A5) (catalogue number: 9246, RRID:AB_2267145) and rabbit anti-Phospho-IKKα/β (Ser$^{176/180}$) (16A6) (catalogue number: 2697, RRID:AB_2079382) were purchased from Cell Signaling Technology (CST). Monoclonal mouse anti-IKKβ (10AG2) (catalogue number: NB100-56509) was purchased from Novus Biologicals. Polyclonal rabbit anti-IKKβ (catalogue number: BB-AB0094) and Polyclonal rabbit anti-6XHis (catalogue number: BB-AB0010) were procured from Bio Bharti Life Science. Monoclonal mouse anti-Phospho-Tyrosine (catalogue number: 610000, RRID:AB_397423) was purchased from BD Biosciences or from EMD Millipore (catalogue number: 05–321 X). Monoclonal mouse anti-NEMO was purchased from BD Biosciences (catalogue number: 559675, RRID:AB_397297). Polyclonal rabbit anti-GST (catalogue number: 924801, RRID:AB_2565461), monoclonal mouse anti-HA 11 epitope (catalogue number: 901502, RRID:AB_2565007) and mouse anti-Tubulinβ3 (catalogue number: 657402, RRID:AB_2562570) were purchased from BioLegend. Polyclonal rabbit anti-IKKα/β (H-470) (catalogue number: sc7607, RRID:AB_675667) was purchased from Santa Cruz Biotechnology (has been discontinued) and mouse anti-Phospho-Serine Q5 (catalogue number: 37430) was from Qiagen. Horseradish peroxidase-conjugated secondary antibodies against rabbit (catalogue number: 31460, RRID:AB_228341) and mouse IgG (catalogue number: 31430, RRID:AB_228307) were Pierce antibodies purchased from Thermo Scientific. Adenosine-5'-Triphosphate (ATP), Disodium trihydrate (catalogue number: A-081–25) was obtained from Gold Biotechnology and Adenosine Diphosphate (ADP; Reference number: V916A) used was provided with the ADP-Glo Kinase assay kit (catalogue number: V6930) from Promega. Recombinant murine TNF-α used was obtained from Roche. Oligonucleotides were purchased from IDT or GCC BioTech. Restriction enzymes and T4 DNA ligase were purchased from New England Biolabs.

Sf9 cells (Gibco Sf-900 III SFM) for expression of proteins using the baculovirus system were procured from Thermo Fisher Scientific (catalogue number: 12659017) and were maintained in media Gibco Sf-900 III SFM, Thermo Fisher Scientific (catalogue number: 12658019). Cells were grown and maintained as per the manufacturer's protocol.

IKK2 knock out (KO) mouse embryonic fibroblast (MEF) cells, generated from E12.5 *ikk2⁻/⁻* mouse embryos, were a kind gift from Prof. Inder Mohan Verma, Salk Institute (*Li et al., 1999*). Immortalized MEF cells were grown in DMEM supplemented with 10% heat-inactivated bovine calf serum, penicillin (100 units/ml), streptomycin (100 μg/ml), and L-glutamine (1%) at 5% $CO_2$ and 37 °C. Authenticity of MEF cells was carefully assessed by observing their characteristic spindle shape morphology and their conformity to 3T3 fibroblasts, and by observing contact inhibition properties. The cells were also routinely assessed for the presence/absence of IKK1 and IKK2, and IκBα degradation dynamics in response to mouse TNFα treatment using immunoblotting (*Polley et al., 2016*). Cells were tested negative for mycoplasma contamination. Mycoplasma contamination/infection leads to the degradation of IκBα protein and NF-κB activation even in the absence of any known external signaling cues (i.e. control untreated cells) and thus is easily recognized and could be further tested. In summary, cells were carefully maintained as adherent monolayer cultures using the 3T3 protocol, and cells not conforming to ideal phenotypic or functional NF-κB signaling characteristics were discarded.

## Protein expression and purification

### IKK2 (WT and other constructs)

Full-length human IKK2 cDNA was a gift from the laboratory of M. Karin (School of Medicine, UC, San Diego) to GG lab. The FL hIKK2 EE construct has been previously described (*Polley et al., 2013*). IKK2Δ700 EE truncate was subcloned into the *BamHI* and *NotI* restriction sites of the pFastBac HTB vector such that it is in frame with the N-terminal hexa-histidine tag followed by TEV protease recognition sequence. For the site-directed mutagenesis, all substitutions of the mutants, namely IKK2 AA, IKK2 Y169F, and IKK2 K44M, were performed within the pFastBac HTB vector harboring FL IKK2 WT. Wild-type and mutant IKK2 proteins were purified following a published protocol (*Polley et al., 2013*). Sf9 cells (Gibco Sf9 cells in Sf-900 III SFM) at a density of 1.5–2 million/ml were infected with a high titer viral stock (generally stage P3 or beyond) at a ratio (usually around 1:100) optimized for a high expression level through a pilot experiment and cultured for 48–60 hr upon infection. Cells were harvested and lysed by sonication in a buffer containing 25 mM Tris-HCl pH 8.0, 200 mM NaCl, 10% Glycerol, 5 mM β-Mercaptoethanol (β-ME), 10 mM Imidazole, 20 mM β-glycerophosphate, 10 mM NaF, 1 mM sodium orthovanadate, and 1 X Protease Inhibitor Cocktail (PIC, Sigma). The lysate was clarified by centrifugation at 20,000 x $g$ for 45 min at 4 °C and incubated with Ni-NTA agarose resin (QIAGEN, catalogue number: 30230) pre-equilibrated with lysis buffer for 2 hr at 4 °C. The resin was subsequently washed with lysis buffer containing 30 mM Imidazole. Protein was eluted with lysis buffer containing 250 mM Imidazole. Protein fractions were analyzed on SDS-PAGE, pooled together, concentrated, and either flash frozen or subjected to size-exclusion chromatography on Superdex 200 Increase 10/300 GL (Cytiva) column equilibrated with 20 mM Tris-HCl pH 8.0/25 mM HEPES pH 7.9, 200 mM NaCl, 10% Glycerol and 2 mM DTT. To produce untagged IKK2, protein fractions eluted from the Ni-NTA resin of desired purity were treated with TEV-protease at a ratio of 50:1 (weight by weight, IKK2: TEV) at 4 °C for 10 hr or longer in the presence of 1 mM of EDTA. The resulting protein mix was then loaded onto the SEC as described above. To prepare autophosphorylated IKK2, the fractions with the desired purity level were treated with 0.5–1 mM of ATP, 10 mM of MgCl$_2$ and phosphatase inhibitors and the reaction mixture was incubated at 27 °C for 1 hr. Resulting reaction mixture was concentrated in a YM30-concentrator and loaded onto a Superdex 200 Increase 10/300 GL column. Elution fractions of the size-exclusion chromatography were analyzed on an SDS-PAGE, and peak fractions of desired quality were pooled together and concentrated using a YM30 Centriprep (Millipore) or Centricon devices (Millipore) depending upon the volume of the sample, and flash frozen in liquid nitrogen immediately.

### IκBα (WT, mutant and other constructs)

Full-length human IκBα was subcloned into the *BamHI* and *SalI* restriction sites of the pET24d vector such that it is in frame with the N-terminal hexa-histidine tag followed by a TEV-protease cleavage site. All substitutions for each mutation, namely S32A/E and S36A/E, as well as S32A,S36A were made through site-directed mutagenesis performed within the pGEX4T-2 vector harboring GST-IκBα (1-54) WT. His-tagged full-length and GST-tagged deletion (1-54) constructs of IκBα (both the wild-type and Ser32Ala, Ser36Ala, Ser32Glu, Ser36Glu, Ser32Ala,Ser36Ala single and double-mutant versions) were expressed in *E. coli* Rosetta2 (DE3). The cells at an attenuance of 0.6 were induced with 0.3 mM IPTG and cultured overnight at 16 °C. The cell pellet was resuspended in 40 ml of Lysis Buffer (25 mM Tris pH 7.5, 200 mM NaCl, 10 mM imidazole, 10% Glycerol, 5 mM β-mercaptoethanol, and 1 mM PMSF). The cells were lysed on ice with a sonicator. The lysate was clarified by centrifugation at 15,000 x $g$ for 1 hr at 4 °C and mixed with 2–3 ml slurry of Ni-NTA Agarose resin that was equilibrated with lysis buffer on a rotary mixer for 2 hr in a 4 °C room. The resin was washed with the lysis buffer containing 500 mM NaCl (high-salt wash) for at least 50 column volumes to remove non-specifically bound proteins. Subsequently, the resin was thoroughly washed with the lysis buffer containing 20 mM imidazole. His-tagged IκBα protein was eluted under gravity flow, using elution buffer (the lysis buffer containing 250 mM imidazole), and elution fractions of ~1 ml volume were collected. For GST-tagged proteins, Glutathione Sepharose resin (Cytiva, catalogue number: 17-5279-01) was used, and the elution was performed with the help of reduced glutathione. The elution fractions containing quality proteins were subjected to size-exclusion chromatography (SEC) on a HiLoad 16/600 pg Superdex 200 column (Cytiva) in 25 mM Tris pH 7.5, 100 mM NaCl, 5% glycerol, and 5 mM DTT. The peak fractions were concentrated, aliquoted, and stored at –80 °C.

## NEMO FL WT

Full-length human NEMO was subcloned into the *EcoRI* and *SalI* restriction sites of the pET24d vector such that it is in frame with the N-terminal hexa-histidine tag. His-tagged FL NEMO WT was expressed in *E. coli* Rosetta2 DE3 cells. The cells were grown at 37 °C in LB medium containing 50 µM ZnCl$_2$ (Millipore) to an attenuance of 0.3–0.4 and cooled down to 20 °C. Upon addition of 0.3 mM IPTG, the culture was further incubated for 14–16 hr at 20 °C. The culture post-induction was cooled down to 4 °C before harvesting cells by centrifugation at 5000 rpm at 4 °C. The cell pellet was resuspended in 40 ml (per liter of culture) of Lysis Buffer 25 mM Tris pH 7.5, 200 mM NaCl, 10 mM imidazole, 10% Glycerol, 5 mM β-mercaptoethanol, 20 µM ZnCl$_2$, 0.5 M urea (as an osmolyte). The rest of the steps performed were similar to purification with Ni-NTA described above. The main elution fractions were subjected to size-exclusion chromatography on a HiLoad 16/600 pg Superdex 200 column (Cytiva) in 25 mM HEPES, 100 mM NaCl, 5% glycerol, and 5 mM β-mercaptoethanol buffer. The cleaner peak fractions were stored, pooled, concentrated, and stored at –80 °C.

## Kinase assay with purified proteins

The presented kinase assay figures are ideal representatives of a minimum of three similar experiments, unless otherwise noted. For all in vitro kinase assays, a master mix was prepared just prior to reaction start to minimize pipetting and other errors. Typically, 50–100 ng (unless otherwise noted) of purified kinase was used in each kinase reaction of 20 µl volume. 0.5–1 µg of IκBα substrate (full-length or GST-tagged 1–54) was incubated with the kinase for 30 min or indicated time periods at 27 °C in the presence of 20 µM ATP spiked with 0.1 µCi of γ-P$^{32}$-ATP in case of radioactive assay or 50–100 µM ATP in case of Western blot assays in a reaction buffer containing 20 mM HEPES pH 7.5, 100 mM NaCl, 15 mM MgCl$_2$, 2 mM DTT, 0.2 mM Na$_3$VO$_4$, 10 mM NaF, 20 mM β-Glycerophosphate. Reaction was stopped by the addition of 1X-Laemmli buffer and heating the samples to 95 °C for 5 min. Samples (half of the total reaction mixture) were resolved on 8 or 10% SDS-PAGE gels. For radioactive assays, the gels were quickly stained and de-stained by Coomassie staining protocol to confirm loading equivalencies. Typically, the substrate band(s) were monitored for this purpose as kinase band(s) were not visible in quick Coomassie staining protocol. For radioactive kinase assays with the IKK2 K44M mutant, a higher amount of kinase was used (0.5–1 µg), and the kinase band was visible following the same staining/de-staining protocol (*Figure 3C* and S3E, Fig. S1A). The gels were subsequently dried and analyzed using Typhoon phosphor imager or by exposing them to Kodak films. For western blot analyses, proteins from the SDS-PAGE were electro-transferred onto nitro-cellulose/PVDF membrane and probed with desired antibodies. Phosphorylated tyrosine and serine residues were detected using monoclonal antibodies against phospho-tyrosine and phospho-serine. Specific phosphorylation on S32/S36 of IκBα was detected using a monoclonal antibody against IκBα phosphorylated at S32/36, and phosphorylation at the activation loop serines of IKK2 was detected using a monoclonal antibody against IKK2 phosphorylated at S177/S181.

For the assays with inhibitors, IKK2 or IKK2:IκBα were mixed with the respective inhibitors in the kinase assay buffer at concentrations indicated in the respective figures for 30 min at room temperature - prior to the addition of ATP or γ-P$^{32}$-ATP. Upon addition of ATP, reactions were allowed to proceed for indicated time periods prior to analysis by autoradiography or immunoblotting as described above.

## IP-kinase assay

HA-tagged IKK2 wildtype and mutant enzymes were cloned into pBABE-Puro retroviral vector. Immortalized *ikk2*$^{-/-}$ mouse embryonic fibroblast 3T3 cells were reconstituted by retroviral transduction with HA-tagged WT or mutant pBABE-IKK2 constructs following a protocol as described in *O'Dea et al., 2008*. Stable expression of IKK2 was confirmed by immunoblotting. Freshly plated cells were stimulated with recombinant murine TNF-α (Roche) when cells reached confluency of ~70–80%. Whole cell extract was prepared and pre-cleared by incubation with Protein A agarose bead for an hour. IKK complex was immunoprecipitated from whole cell extracts using a NEMO-specific antibody (BD) and Protein A agarose (Upstate) for 2 hr at 4 °C. Immunoprecipitated complex was washed extensively, including a final washing step with the kinase assay buffer devoid of ATP and incubated with GST-IκBα (1-54) and 20 µM ATP spiked with 5 µCi of γ-$^{32}$P-ATP for 30 min at 30 °C in kinase assays. Reactions were stopped by the addition of 2X-Laemmli SDS buffer and heating at 95 °C for 5 min prior to

resolving on a 12% SDS PAGE. Extent of substrate phosphorylation was measured by cutting out the appropriate area of the gel containing the substrate and exposing it to phosphor-imager plates (Typhoon, Cytiva) after drying. For control of the amount of IKK2 in each reaction, IKK2 was transferred from gel onto nitrocellulose membrane and probed with anti-IKK1/2 antibody. These assays were performed twice, and representative data images are shown.

## Differential scanning calorimetry (DSC)

IKK2Δ700 EE protein was treated with 1 mM ATP or ADP in an in vitro kinase assay for 1 hr at 30 °C and subjected to size exclusion chromatography on a Superdex 200 column pre-equilibrated with 20 mM Na-Phosphate, pH 7.4, 200 mM NaCl and 1 mM TCEP. Peak fractions were collected, and samples were subjected to DSC run at 1 mg/ml concentration on a N-DSC II differential scanning calorimeter (Calorimetry Sciences Corp, Provo, UT) at a scanning rate of 1 K/min under 3.0 atm of pressure. DSC data were analyzed using CpCalc software provided by the DSC manufacturer.

## Sample preparation for LC-MS/MS analyses for tyrosine phosphorylation detection

A kinase assay reaction was set up with about 20 µg of FL IKK2 WT in the presence of 100 µM ATP for about an hour at 27 °C. SDS to a final concentration of 0.1% was added, and the reaction was incubated at 30 °C for 30 min. The sample buffer was exchanged to 50 mM Ammonium bicarbonate (Sigma), 8 M urea (Merck) and 2.5 mM TCEP using a 10 kDa concentrator (EMD Millipore), and the sample was further incubated at 37 °C for 45 min. Next, 20 mM iodoacetamide (IAA) was added (final concentration) to the sample and incubated in the dark at room temperature for an hour. After this step, 5 mM Dithiothreitol (DTT; Promega, MS grade) was added and incubated at room temperature for an hour. The sample was then passed through a protein-desalting column PD SpinTrap (Cytiva) to get rid of urea. GluC protease in the final ratio of protease: protein that is 1:50 (w/w) was then added and incubated for an hour at 37 °C. Next, Trypsin Gold (Promega, MS grade) was added to the sample at a final ratio of protease: protein that is 1:100 (w/w) and incubated at 37 °C for 2 hr. Finally, Trypsin Gold was added in the final ratio of 1:50 (w/w) to the sample for overnight incubation at 37 °C. On the following day, 0.1% of formic acid (final concentration) was added to inactivate trypsin and incubated for 5 min at room temperature. The samples were quickly frozen at –80 °C and then dried out using a vacuum evaporator (Savant RVT5105, Thermo Fisher Scientific). The trypsinized sample was then resuspended in 0.5% TFA and 5% ACN solution. The sample was passed through the C18 spin column (Pierce, Thermo Fisher Scientific) by following the manual. Finally, the elution was performed in 70% ACN and 0.1% formic acid and then the elute was dried out using the vacuum evaporator.

## Preparation of FL IKK2 K44M for mass spectrometry to check purity

About 25 µg of purified protein was taken and denatured at 90 °C for 10 min. 3 mM Tris (2-carboxyethyl) phosphine (TCEP) (GoldBiochem) was then added, and the reaction was incubated for 45 min at 37 °C. Next, 20 mM iodoacetamide (IAA) was added (final concentration) to the sample tube and incubated in the dark at room temperature for an hour. After this step, 5 mM Dithiothreitol (DTT) (Promega, MS grade) was added and incubated at room temperature for an hour to neutralize the effect of excess IAA. Next, Trypsin Gold (Promega, MS grade) was added to the sample at a final ratio of protease: protein that is 1:100 (w/w) and incubated at 37 °C for 2 hr. Finally, Trypsin Gold was added in the final ratio of 1:50 (w/w) to the sample for overnight incubation at 37 °C. On the following day, 0.1% of formic acid (final concentration) was added to inactivate trypsin and incubated for 5 min at room temperature. The samples were quickly frozen at –80 °C and then dried out using a vacuum evaporator (Savant RVT5105, Thermo Fisher Scientific). The trypsinized sample was then resuspended in 0.5% TFA and 5% ACN solution. The sample was passed through the C18 spin column (Pierce, Thermo Fisher Scientific) following the manufacturer's protocol. Finally, the elution was performed in 70% ACN and 0.1% formic acid and the elute was dried out using the vacuum evaporator.

## LC-MS/MS analysis

For analyses on an Orbitrap platform, peptides were resuspended in 5% (v/v) formic acid and sonicated for 5 min. Samples were analyzed on Orbitrap Exploris 240 mass spectrometer (Thermo Fisher Scientific) coupled to a nanoflow LC system (Easy nLC II, Thermo Fisher Scientific). Peptides were

loaded onto a PepMap RSLC C18 nanocapillary reverse phase HPLC column (75 µm×15 cm; 3 µm; 100 Å) and separated using a 60 min linear gradient of the organic mobile phase [5% Acetonitrile (ACN) containing 0.2% formic acid and 90% ACN containing 0.2% formic acid]. For identification of peptides, the raw data was analyzed on the MaxQuant proteomics computational platform (Ver. 1.6.8; *Cox and Mann, 2008*) and searched against UniProt amino acid sequences (Uniprot Reference Proteome ID: UP000829999; Uniprot ID for IKK2: O14290). MaxQuant used a decoy version of the specified database to adjust the false discovery rates for proteins and peptides below 1%. The search parameters included constant modification of cysteine by carbamidomethylation, phosphorylation (STY) as a variable modification and enzyme specificity as trypsin. The iBAQ option was selected to compute the abundance of the proteins.

## ESI-MS analyses of ADP

A 50 µM solution of ADP was prepared for analysis by Electrospray Ionisation Mass Spectrometry (ESI-MS) in a Xevo G2-XS QToF mass spectrometer (Waters Corporation) using capillary at 3kV, with voltage parameters of sampling cone at 40 V and source offset at 80 V. The source temperature was set at 100 °C and desolvation temperature at 40 °C. The gas flow rates were as follows: Cone gas flow rate at 50 l/hr and desolvation gas flow rate at 400 l/hr. The sample was passed at a 5 µl/min flow rate. The MS data was acquired for a mass range of 50–2000 m/z, and the acquisition time was 1 min.

## Radioactive phospho-transfer assay

Phospho-transfer assays were performed in a wide range of concentrations of IKK2, IκBα, and ATP. Typically, a 10 µM concentration of purified IKK2 protein was autophosphorylated in the presence of 100–200 nM of $\gamma^{32}$P-ATP (PerkinElmer) in kinase assay buffer for 1 hr at 30 °C. 70 µl of the reaction mixture was passed through a Micro Bio-Spin Bio-Gel P-30 spin column (Bio-Rad; 40 kDa MWCO) equilibrated with 20 mM HEPES pH 7.9, 100 mM NaCl, 0.2 mM Na$_3$VO$_4$, 10 mM NaF, 20 mM β-Glycerophosphate, 2 mM DTT, 2 mM EDTA and 1 X protease inhibitor cocktail. Eluted protein was subjected to another round of buffer exchange on a Bio-Gel P-30 column equilibrated with the same buffer mentioned above, devoid of EDTA and containing 15 mM MgCl$_2$. Phosphorylated IKK2 thus obtained was used in the phosphotransfer reaction. 2–3 µM of Phosphorylated IKK2 was incubated with 10–20 µM of WT or mutant IκBα protein at 30 °C in kinase assay buffer in a total volume of 20 µl for different time periods as indicated in *Figures 5A and 6A*. Presented figure is a representative of a minimum of three similar assays unless otherwise noted. This assay was performed twice with the mutant IκBα protein (*Figure 6A*). The reactions were stopped by the addition of 10 µl of 4 X Laemmli SDS-buffer containing 100 mM EDTA to samples and heating at 95 °C for 5 min. It was observed that the addition of SDS and EDTA was sufficient to stop the reaction without heating. Phospho-transfer signal was not dependent upon heating of the reaction mixture. Different concentrations of ADP or ATP were added to the assay to assess the role of these nucleotides in phosphotransfer efficiency. Routinely, 10–15 µl of reaction products were resolved on 10% SDS PAGE, and radiolabeled samples were quantified by exposing the dried gel to phosphor-imager plates scanned on a Typhoon scanner. For the phosphotransfer reaction described in *Figure 5A*, 10 µM purified IKK2 was autophosphorylated with 100 nM of $\gamma^{32}$P-ATP in the presence of 40 µM of cold ATP. Each concentration mentioned above is the final concentration.

## Phospho-transfer assay with cold ATP

~10 µM of purified FL IKK2WT was mixed with 100 µM of cold ATP in 600 µl volume of kinase assay buffer (containing 1 X protease inhibitor cocktail) for one hour at 30 °C. From the resulting reaction mixture, 500 µl was loaded onto a Superdex 200 increase 10/300 GL (24 ml bed volume, Cytiva) SEC column to separate autophosphorylated FL IKK2WT from excess ATP. SEC was performed in buffer containing 20 mM HEPES, 100 mM NaCl, 20 mM β-glycerophosphate, 10 mM NaF, 0.2 mM Na$_3$VO$_4$ and 2 mM DTT. The elution fractions containing phosphorylated IKK2 were incubated with FL IκBαWT in the presence of different concentrations of ATP or ADP at different time points (as mentioned in *Figure 5E*) at 30 °C. The reactions were stopped by the addition of 4 X Laemmli SDS dye and heating at 95 °C for 5 min and prior to resolving on a 10% SDS-PAGE. Immunoblotting was performed using monoclonal antibodies against phospho-tyrosine and phospho Ser32/36 IκBα to detect phosphorylations. This assay was performed twice, and the representative figure is shown (*Figure 5E*).

## Molecular dynamics simulation

The kinase domain structure of IKK2 (PDB ID: 4E3C), spanning residues 16–310, was obtained from the PDB database (*Berman et al., 2000*). We created unphosphorylated forms (at Ser177 & Ser181) and two phosphorylated forms (phosphorylated at Ser177 & Ser181 and at Tyr169 & Ser177 & Ser181) of IKK2 using UCSF CHIMERA software (*Pettersen et al., 2004*). These structures underwent molecular dynamics (MD) simulations using the GROMACSv2023.1 simulation package (*Abraham et al., 2015*), enabling a comparative analysis of their structural and energetic stability. The Charmm36 force field was applied to generate coordinates and topology files for the IKK2 structures (*Huang and MacKerell, 2013*). For simulation, a cubic box was defined and filled with TIP3 water molecules (*Jorgensen et al., 1983*). Sodium and chloride ions were added to neutralize the three systems. The system underwent two-stage minimization using steepest-descent (*Nocedal and Wright, 1999*) and conjugate-gradient algorithms (*Straeter, 1971*). Equilibration was carried out under NVT (constant particle number, volume, and temperature) and NPT (constant particle number, pressure, and temperature) conditions for 200 ps at 300 K and 1 atm pressure. MD simulations were conducted for 200 ns for each protein. To assess global structural changes, Root Mean Square Deviation (RMSD) analysis was employed, while Root Mean Square Fluctuation (RMSF) analysis revealed the overall and local motion/flexibility of the proteins. The *gmx_energy* module was utilized to compute energy during simulation, and *gmx_sasa* was employed to determine the solvent accessible surface area (SASA) of the protein. To understand the correlations of the atomic motions in different regions, the dynamic cross-correlation analysis was carried out. The initial step of dynamic cross-correlation analysis was to construct the covariance matrix, which examines the linear relationship of atomic fluctuations for individual atomic pairs. The covariance matrix was calculated using the *gmx_covar* tool. Dynamic cross-correlation matrices (DCCM) were calculated from the $C_\alpha$-trace covariance matrix principal components. Results are displayed as color-coded, with a value of −1 indicating completely anti-correlated motions and a value of +1 indicating completely correlated motions.

## Molecular docking analysis

The initial (0th ns) and terminal (200th ns) structures from the simulations of unphosphorylated (UnP-IKK2), double phosphorylated (p-IKK2), and triple phosphorylated (P-IKK2) forms were subjected to molecular docking analysis using ATP and ADP as ligands, allowing for comparison of binding affinities. LeDock (*Liu and Xu, 2019*) and GOLD software were employed for ligand-protein docking. IKK2: ATP docking poses were compared with available kinase:ATP complexes, for example PKA: ATP complex (PDB ID: 1ATP) by calculating the ligand root mean square deviation (RMSD). The pose that resembled most with the pose found in PDB ID 1ATP was chosen for further analysis. Similarly, IKK2: ADP docking poses were also compared, where IRE1: ADP (PDB ID: 2RIO) complex was chosen for comparison, and the pose closest to that in PDB ID: 2RIO was chosen for further analysis. Finally, the best selected poses were rescored using AutoDock Vina to represent the predicted binding affinities.

For assessing the ability of P-IKK2 to accommodate ATP or ADP, the MD-simulation-derived P-IKK2 structure was superimposed on the p-IKK2 structure bound to ATP/ADP and compared as described.

## Estimation of binding free energy

Following the identification of the best docking pose, another round of molecular dynamics (MD) simulation was conducted using GROMACS for 10 ns. 50 intermediate complex structures were extracted from each simulation run. The binding free energy (ΔG) of these intermediate structures was calculated using the MM-PBSA (Molecular Mechanics Poisson-Boltzmann Surface Area) method.

## Acknowledgements

We thank Tony Hunter (Salk Institute) and Kaushik Biswas (Bose Institute) for commenting on the work, and members of Polley lab for their continuous support. This work was supported by an Intermediate Fellowship from the DBT Wellcome Trust India Alliance (IA/I/15/1/501852) & intramural funding from Bose Institute to SP, NIH grant to GG (AI163327), and American Cancer Society grant RSG-08-287-01-GMC and California Metabolic Research Foundation support to Biochemistry research at SDSU to TH. PB was supported by the graduate research fellowship from Bose Institute. SC acknowledges

CSIR-IICB for infrastructure support. AC acknowledges ICMR for post-doctoral Research Associateship [BMI/11(55)2022].

## Additional information

### Funding

| Funder | Grant reference number | Author |
|---|---|---|
| Wellcome Trust DBt India Alliance | IA/I/15/1/501852 | Smarajit Polley |
| Indian Council of Medical Research | BMI/11(55)2022 | Ankur Chaudhuri |
| National Institutes of Health | NIH AI163327 | Gourisankar Ghosh |
| American Cancer Society | RSG-08-287-01-GMC | Tom Huxford |

The funders had no role in study design, data collection and interpretation, or the decision to submit the work for publication. For the purpose of Open Access, the authors have applied a CC BY public copyright license to any Author Accepted Manuscript version arising from this submission.

### Author contributions

Prateeka Borar, Tapan Biswas, Data curation, Validation, Investigation, Visualization, Methodology, Writing – review and editing; Ankur Chaudhuri, Data curation, Software, Formal analysis, Validation, Investigation, Visualization, Performed computational analyses; Pallavi T Rao, Data curation, Investigation, Visualization, Performed LC-MS analyses; Swasti Raychaudhuri, Resources, Data curation, Supervision, Supervised LC-MS analyses; Tom Huxford, Writing – review and editing; Saikat Chakrabarti, Resources, Data curation, Software, Formal analysis, Supervision, Methodology, Designed and supervised computational analyses; Gourisankar Ghosh, Resources, Supervision, Investigation, Methodology, Writing – review and editing; Smarajit Polley, Conceptualization, Resources, Data curation, Supervision, Validation, Investigation, Visualization, Methodology, Writing – original draft, Project administration, Writing – review and editing

### Author ORCIDs

Prateeka Borar ⓘ https://orcid.org/0009-0001-3583-2179
Ankur Chaudhuri ⓘ https://orcid.org/0000-0002-8328-6643
Swasti Raychaudhuri ⓘ https://orcid.org/0000-0003-4091-7991
Tom Huxford ⓘ http://orcid.org/0000-0002-1939-7373
Gourisankar Ghosh ⓘ https://orcid.org/0000-0001-6311-7351
Smarajit Polley ⓘ https://orcid.org/0000-0003-3494-5267

Reviewer #1 (Public review): https://doi.org/10.7554/eLife.98009.4.sa1
Reviewer #2 (Public review): https://doi.org/10.7554/eLife.98009.4.sa2
Reviewer #3 (Public review): https://doi.org/10.7554/eLife.98009.4.sa3
Author response https://doi.org/10.7554/eLife.98009.4.sa4

## Additional files

### Supplementary files

Supplementary file 1. MaxQuant output file (Excel file) of LC MS/MS analysis for FL IKK2 K44M.

Supplementary file 2. Number of unique peptides, spectral counts, and iBAQ values from mass spectrometric analysis of FL IKK2 K44M shown in 3D scatter plot in *Figure 1—figure supplement 1D*.

Supplementary file 3. Details for the 200ns energy trajectory of three differently phosphorylated states of IKK2 - UnP-IKK2, p-IKK2, and P-IKK2 shown in *Figure 4A*.

Supplementary file 4. Details for the 200ns of RMSD of three differently phosphorylated states of IKK2 - UnP-IKK2, p-IKK2, and P-IKK2 shown in *Figure 4B*.

Supplementary file 5. Details for the 200ns of RMSF of three differently phosphorylated states of IKK2 - UnP-IKK2, p-IKK2, and P-IKK2 shown in *Figure 4—figure supplement 1C*.

Supplementary file 6. Ratios of the average intensity of the phosphorylated protein with respect to the intensity of total protein for each lane plotted in *Figure 5—figure supplement 1A*.

Supplementary file 7. Data points for the Differential Scanning Calorimetry experiment shown in *Figure 4—figure supplement 1A*.

MDAR checklist

### Data availability

All computational data supporting the findings of this study are available in a repository on Zenodo (https://zenodo.org/records/15309725).

The following dataset was generated:

| Author(s) | Year | Dataset title | Dataset URL | Database and Identifier |
|---|---|---|---|---|
| Borar P, Biswas T, Chaudhuri A, Rao PT, Raychaudhuri S, Huxford T, Chakrabarti S, Ghosh G, Polley S | 2025 | Dual-specific autophosphorylation of kinase IKK2 enables phosphorylation of substrate I-kappaBalpha through a phosphoenzyme intermediate | https://doi.org/10.5281/zenodo.15309725 | Zenodo, 10.5281/zenodo.15309725 |

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
