## [Editor Report · eLife Assessment]

This study presents **fundamental** findings that could redefine the specificity and mechanism of action of the well-studied Ser/Thr kinase IKK2 (a subunit of inhibitor of nuclear factor kappa-B kinase (IKK) that propagates cellular response to inflammation). **Solid** evidence supports the claim that IKK2 exhibits dual specificity that allows tyrosine autophosphorylation and the authors further show that auto-phosphorylated IKK2 is involved in an unanticipated relay mechanism that transfers phosphate from an IKK2 tyrosine onto the IκBα substrate. The findings are a starting point for follow-up studies to confirm the unexpected mechanism and further pursue functional significance.

---

## [Referee Report · Reviewer #1 (Public review)]

IKK is the key signaling node for inflammatory signaling. Despite the availability of molecular structures, how the kinase achieves its specificity remains unclear. This paper describes a dynamic sequence of events in which autophosphorylation of a tyrosine near the activate site facilitates phosphorylation of the serine on the substrate via a phosphor-transfer reaction. The proposed mechanism is conceptually novel in several ways, suggesting that the kinase is dual specificity (tyrosine and serine) and that it mediates a phospho-transfer reaction. While bacteria contain phosphorylation-transfer enzymes, this is unheard of for mammalian kinases. However, what the functional significance of this enzymatic activity might remain unaddressed.

The revised manuscript adequately addresses all the points I suggested in the review of the first submission.

---

## [Referee Report · Reviewer #2 (Public review)]

The authors investigate the phosphotransfer capacity of Ser/Thr kinase IκB kinase (IKK), a mediator of cellular inflammation signaling. Canonically, IKK activity is promoted by activation loop phosphorylation at Ser177/Ser181. Active IKK can then unleash NF-κB signaling by phosphorylating repressor IκBα at residues Ser32/Ser26. Noting the reports of other IKK phosphorylation sites, the authors explore the extent of autophosphorylation.

Semi-phosphorylated IKK purified from Sf9 cells, exhibits the capacity for further autophosphorylation. Anti-phosphotyrosine immunoblotting indicated unexpected tyrosine phosphorylation. Contaminating kinase activity was tested by generating a kinase-dead K44M variant, supporting the notion that the unexpected phosphorylation was IKK-dependent. In addition, the observed phosphotyrosine signal required phosphorylated IKK activation loop serines.

Two candidate IKK tyrosines were examined as the source of the phosphotyrosine immunoblotting signal. Activation loop residues Tyr169 and Tyr188 were each rendered non-phosphorylatable by mutation to Phe. The Tyr variants decreased both autophosphorylation and phosphotransfer to IκBα. Likewise, Y169F and Y188F IKK2 variants immunoprecipitated from TNFa-stimulated cells also exhibited reduced activity in vitro.

The authors further focus on Tyr169 phosphorylation, proposing a role as a phospho-sink capable of phosphotransfer to IκBα substrate. This model is reminiscent of the bacterial two-component signaling phosphotransfer from phosphohistidine to aspartate. Efforts are made to phosphorylate IKK2 and remove ATP to assess the capacity for phosphotransfer. Phosphorylation of IκBα is observed after ATP removal, although there are ambiguous requirements for ADP.

Strengths:

Ultimately, the authors draw together the lines of evidence for IKK2 phosphotyrosine and ATP-independent phosphotransfer to develop a novel model for IKK2-mediated phosphorylation of IκBα. The model suggests that IKK activation loop Ser phosphorylation primes the kinase for tyrosine autophosphorylation. With the assumption that IKK retains the bound ADP, the phosphotyrosine is conformationally available to relay the phosphate to IκBα substrate. The authors are clearly aware of the high burden of evidence required for this unusual proposed mechanism. Indeed, many possible artifacts (e.g., contaminating kinases or ATP) are anticipated and control experiments are included to address many of these concerns. The analysis hinges on the fidelity of pan-specific phosphotyrosine antibodies, and the authors have probed with two different anti-phosphotyrosine antibody clones. Taken together, the observations are thought-provoking, and I look forward to seeing this model tested in a cellular system.

Weaknesses:

Multiple phosphorylated tyrosines in IKK2 were apparently identified by mass spectrometric analyses. LC-MS/MS spectra are presented, but fragments supporting phospho-Y188 and Y325 are difficult to distinguish from noise. It is common to find non-physiological post-translational modifications in over-expressed proteins from recombinant sources. Are these IKK2 phosphotyrosines evident by MS in IKK2 immunoprecipitated from TNFa-stimulated cells? Identifying IKK2 phosphotyrosine sites from cells would be especially helpful in supporting the proposed model.

---

## [Referee Report · Reviewer #3 (Public review)]

Summary:

The authors investigate the kinase activity of IKK2, a crucial regulator of inflammatory cell signaling. They describe a novel tyrosine kinase activity of this well-studied enzyme and a highly unusual phosphotransfer from phosphorylated IKK2 onto substrate proteins in the absence of ATP as a substrate.

Strengths:

The authors provide an extensive biochemical characterization of the processes with recombinant protein, western blot, autoradiography, protein engineering and provide MS data now.

Weaknesses:

The identity and purity of the used proteins has improved in the revised work. Since the findings are so unexpected and potentially of wide-reaching interest - this is important. Similar specific detection of phospho-Ser/Thr vs phospho-Tyr relies largely on antibodies which can have varying degrees of specificity. Using multiple antibodies and MS improves the quality of the data.

---

## [Author Response]

The following is the authors’ response to the previous reviews

**Public Reviews:**

**Reviewer #1 (Public review):**
IKK is the key signaling node for inflammatory signaling. Despite the availability of molecular structures, how the kinase achieves its specificity remains unclear. This paper describes a dynamic sequence of events in which autophosphorylation of a tyrosine near the activate site facilitates phosphorylation of the serine on the substrate via a phosphor-transfer reaction. The proposed mechanism is conceptually novel in several ways, suggesting that the kinase is dual specificity (tyrosine and serine) and that it mediates a phospho-transfer reaction. While bacteria contain phosphorylation-transfer enzymes, this is unheard of for mammalian kinases. However, what the functional significance of this enzymatic activity might remain unaddressed.The revised manuscript adequately addresses all the points I suggested in the review of the first submission.

Response: Authors thank the reviewer for their valuable comments and constructive criticisms for the betterment of the manuscript. We also thank them for appreciating our work. We agree with the reviewer that the functional significance of this particular enzymatic activity of IKK2 is yet to be fully realized.

**Reviewer #2 (Public review):**
The authors investigate the phosphotransfer capacity of Ser/Thr kinase IkB kinase (IKK), a mediator of cellular inflammation signaling. Canonically, IKK activity is promoted by activation loop phosphorylation at Ser177/Ser181. Active IKK can then unleash NF-kB signaling by phosphorylating repressor IkBα at residues Ser32/Ser26. Noting the reports of other IKK phosphorylation sites, the authors explore the extent of autophosphorylation.Semi-phosphorylated IKK purified from Sf9 cells, exhibits the capacity for further autophosphorylation. Anti-phosphotyrosine immunoblotting indicated unexpected tyrosine phosphorylation. Contaminating kinase activity was tested by generating a kinase-dead K44M variant, supporting the notion that the unexpected phosphorylation was IKK-dependent. In addition, the observed phosphotyrosine signal required phosphorylated IKK activation loop serines.Two candidate IKK tyrosines were examined as the source of the phosphotyrosine immunoblotting signal. Activation loop residues Tyr169 and Tyr188 were each rendered non-phosphorylatable by mutation to Phe. The Tyr variants decreased both autophosphorylation and phosphotransfer to IkBα. Likewise, Y169F and Y188F IKK2 variants immunoprecipitated from TNFa-stimulated cells also exhibited reduced activity in vitro.The authors further focus on Tyr169 phosphorylation, proposing a role as a phospho-sink capable of phosphotransfer to IkBα substrate. This model is reminiscent of the bacterial two-component signaling phosphotransfer from phosphohistidine to aspartate. Efforts are made to phosphorylate IKK2 and remove ATP to assess the capacity for phosphotransfer. Phosphorylation of IkBα is observed after ATP removal, although there are ambiguous requirements for ADP.Strengths:Ultimately, the authors draw together the lines of evidence for IKK2 phosphotyrosine and ATP-independent phosphotransfer to develop a novel model for IKK2-mediated phosphorylation of IkBα. The model suggests that IKK activation loop Ser phosphorylation primes the kinase for tyrosine autophosphorylation. With the assumption that IKK retains the bound ADP, the phosphotyrosine is conformationally available to relay the phosphate to IkBα substrate. The authors are clearly aware of the high burden of evidence required for this unusual proposed mechanism. Indeed, many possible artifacts (e.g., contaminating kinases or ATP) are anticipated and control experiments are included to address many of these concerns. The analysis hinges on the fidelity of pan-specific phosphotyrosine antibodies, and the authors have probed with two different anti-phosphotyrosine antibody clones. Taken together, the observations are thought-provoking, and I look forward to seeing this model tested in a cellular system.Weaknesses:Multiple phosphorylated tyrosines in IKK2 were apparently identified by mass spectrometric analyses. LC-MS/MS spectra are presented, but fragments supporting phospho-Y188 and Y325 are difficult to distinguish from noise. It is common to find non-physiological post-translational modifications in over-expressed proteins from recombinant sources. Are these IKK2 phosphotyrosines evident by MS in IKK2 immunoprecipitated from TNFa-stimulated cells? Identifying IKK2 phosphotyrosine sites from cells would be especially helpful in supporting the proposed model.

Authors thank the reviewer for their elaborate comments and constructive criticisms that helped enrich the manuscript. We also thank them for pointing out the critical points in the model. We agree with the reviewer that testing this model in a cellular system is required to bolster this concept. However, an appropriate cellular assay system to investigate and monitor this mode of phosphotransfer is still elusive. We agree with the reviewer’s concerns on the identification of Y188 and Y325 as potential phosphosites. They have been omitted in the current version and relevant changes have been incorporated. IKK2’s tyrosine phosphorylation status in cells is reported earlier. Although we have not analyzed IKK2 from TNFα treated cells in this study, a different study of phospho-status of cellular IKK2 indicated tyrosine phosphorylation (Meyer et al 2013).

**Reviewer #3 (Public review):**
Summary:The authors investigate the kinase activity of IKK2, a crucial regulator of inflammatory cell signaling. They describe a novel tyrosine kinase activity of this well-studied enzyme and a highly unusual phosphotransfer from phosphorylated IKK2 onto substrate proteins in the absence of ATP as a substrate.Strengths:The authors provide an extensive biochemical characterization of the processes with recombinant protein, western blot, autoradiography, protein engineering and provide MS data now.Weaknesses:The identity and purity of the used proteins has improved in the revised work. Since the findings are so unexpected and potentially of wide-reaching interest - this is important. Similar specific detection of phospho-Ser/Thr vs phospho-Tyr relies largely on antibodies which can have varying degrees of specificity. Using multiple antibodies and MS improves the quality of the data.

Authors thank the reviewer for their crisp comments and constructive criticisms that helped improve the manuscript.

**Recommendations for the authors:**

**Reviewer #1 (Recommendations for the authors):**
Generally, the paper is well written, but the first 4 figures are slow going and could be condensed to show the key points, so that reader gets to Figure 6 and 7 which contain the "meat" of the paper.Specific points:Several figures should be quantified and experimental reproducibility is not always clear.I understand that Figure 3 shows that K44M abolishes both S32/26 phosphorylation and tyrosine phosphorylation, but not PEST region phosphorylation. This suggests that autophosphorylation is reflective of its known specific biological role in signal transduction. But I do not understand why "these results strongly suggest that IKK2-autophosphorylation is critical for its substrate specificity". That statement would be supported by a mutant that no longer autophosphorylates, and as a result shows a loss of substrate specificity, i.e. phosphorylates non-specific residues more strongly. Is that the case? Maybe Darwech et al 2010 or Meyer et al 2013 showed this? Later figures seem to address this point, so maybe this conclusion should be stated later in the paper.Page 10: mentions DFG+1 without proper introduction. The Chen et al 2014 paper appears to inform the author's interest in Y169 phosphorylation, or is just an additional interesting finding? Does this publication belong in the Introduction or the Discussion?To understand the significance of Figure 4D, we need a WT IKK2 control: or is there prior literature to cite?This is relevant for the conclusion that Y169 phosphorylation is particularly important for S32 phosphorylation.The cold ATP quenching experiment is nice for testing the model that Y169 functions as a phospho sink that allows for a transfer reaction. However, there is only a single timepoint and condition, which does not allow for a quantitative analysis. Furthermore, a positive control would make this experiment more compelling, and Y169F mutant should show that cold ATP quenching reduces the phosphorylation of IkBa.Note after revision: I thank the authors for addressing these points. The manuscript is thereby improved.

We thank the reviewer for appreciating our efforts in addressing their concerns.

**Reviewer #2 (Recommendations for the authors):**
In the revisions, the authors provide LC-MS/MS spectra for putative phospho-Y325 and phospho-Y188. The details are hard to see at the scale provided, but the fragment ions for pY188 and pY325 peptides are unconvincing. Phospho-Y169, on the other hand, is much more credible. In addition, the revision rebuttal clarifies that Y188 would be packed into a catalytically important core, and Y188F is likely to disrupt the fold. Taken together, it seems doubtful that Y188 is subject to any significant autophosphorylation, and presenting the Y188F data (and discussion) seems like a distraction.

We agree with the reviewer’s concerns on the identification of Y188 and Y325 as potential phosphosites. They have been omitted in the current version and relevant sections in the manuscript text and figures have been edited.